# Longitudinal tracking of neuronal activity from the same cells in the developing brain using Track2p

Jure Majnik[1]*, Manon Mantez[1], Sofia Zangila[1], Stéphane Bugeon[1], Léo Guignard[2], Jean-Claude Platel[1]*†, Rosa Cossart[1]*†‡

[1]Aix-Marseille Université, Inserm, INMED, Turing Center for Living Systems, Marseille, France; [2]Aix-Marseille Université, CNRS, IBDM, UMR7288, Turing Centre for Living Systems, Marseille, France

*For correspondence:
jure.majnik@inserm.fr (JM);
jean-claude.platel@inserm.fr
(J-CP);
rosa.cossart@inserm.fr (RC)

†These authors contributed
equally to this work

‡Lead contact

Competing interest: The authors
declare that no competing
interests exist.

Reviewing Editor: Leopoldo
Petreanu, Champalimaud Center
for the Unknown, Portugal

## eLife Assessment

This **fundamental** study presents a new method for longitudinally tracking cells in two-photon imaging data that addresses the specific challenges of imaging neurons in the developing cortex. It provides **compelling** evidence demonstrating reliable longitudinal identification of neurons across the second postnatal week in mice. The study should be of interest to development neuroscientists engaged in population-level recordings using two-photon imaging.

**Abstract** Understanding cortical circuit development requires tracking neuronal activity across days in the growing brain. While in vivo calcium imaging now enables such longitudinal studies, automated tools for reliably tracking large populations of neurons across sessions remain limited. Here, we present a novel cell tracking method based on sequential image registration, validated on calcium imaging data from the barrel cortex of mouse pups over 1 postnatal week. Our approach enables robust long-term analysis of several hundreds of individual neurons, allowing quantification of neuronal dynamics and representational stability over time. Using this method, we identified a key developmental transition in neuronal activity statistics, marking the emergence of arousal state modulation. Beyond this key finding, our method provides an essential tool for tracking developmental trajectories of individual neurons, which could help identify potential deviations associated with neurodevelopmental disorders.

## Introduction

Early postnatal development in rodents is a period of intense circuit wiring and remodelling at several scales through various major processes that include neuronal growth, synaptogenesis, apoptosis, migration, rise of intracortical connectivity, functional maturation of inhibitory synapses, and the disappearance of transient connectivity schemes (*Cossart and Garel, 2022*; *Molnár et al., 2020*; *Blankenship and Feller, 2010*; *Reh et al., 2020*). Critical periods for various sensory systems open and close during this time, further highlighting the profound reshaping of cortical networks (*Reh et al., 2020*). This developmental period of remodelling is crucial for establishing the functional architecture of the mature cortex. Most importantly, all of these developmental processes are activity-dependent, with collective dynamics playing a critical role in the proper integration of neurons into functional networks (*Wu et al., 2024*). These network dynamics sequentially unfold while relying on different mechanisms and circuits for their generation (*Cossart and Garel, 2022*; *Wu et al., 2024*). The emergence of

**eLife digest** In the weeks and months after birth, the brain experiences rapid growth and restructuring through carefully timed developmental sequences. These processes rely heavily on coordinated neuronal activity for the brain to develop healthily.

Understanding these processes requires recording neural activity from the same cells across different stages of development. However, this has proven technically challenging because both the anatomy and cellular organization of the brain change dramatically during early life.

Two-photon calcium imaging, which uses fluorescent markers to visualize neuronal activity in living animals, has emerged as a powerful method for studying neural circuits. In adult animals, automated techniques can track large neuronal populations across days. However, the rapid brain growth and morphological changes during development make it difficult to track neurons during this stage. Recording from the same developing neurons would provide a more dynamic view of healthy brain maturation and reveal how it diverges in neurodevelopmental conditions.

Majnik et al. investigated whether it was possible to automatically and reliably track the same neurons across consecutive days during early postnatal development of mice. They developed Track2p, an open-source algorithm that can track developing neurons across days using a two-step procedure. It first corrects for brain growth using image registration and then matches neurons across days using this growth-corrected alignment.

Applying Track2p to calcium imaging data from the mouse barrel cortex during the second postnatal week, Majnik et al. found that the algorithm robustly tracked hundreds of neurons despite substantial brain growth. Benchmarking against manually tracked neurons confirmed high accuracy. Analysis of the tracked population's activity properties revealed an increase in overall activity rates and a decrease in firing synchrony. Moreover, around postnatal day 11, neuronal activity patterns shifted from highly synchronized and spatially organized population events to more decorrelated, behavior-dependent firing.

Majnik et al. demonstrate that Track2p overcomes key technical barriers and reveal new principles of early postnatal brain maturation. The tool enables longitudinal analysis of neural circuits, allowing researchers to measure how individual neurons develop over time. Tracking cells across days will be particularly useful for understanding how genetic mutations or environmental factors influence developmental trajectories. Beyond development, Track2p can also examine long-term properties of adult circuits, such as learning or representational stability. Ultimately, applying Track2p to disease models may identify early circuit-level biomarkers of neurodevelopmental disorders.

functional brain circuits during development is therefore a precisely timed choreography, the timing of which is inherently tied to the age of the organism under study.

However, developmental age lacks precision due to significant variations in physical characteristics and growth patterns, even among offspring from the same genetic lineage (*Figure 1*). Hence, longitudinal imaging from the same animal across days is the optimal solution to better capture the evolution of circuit dynamics during development. Additionally, developmental variability extends beyond the organism level, becoming even more pronounced at the single neuron level (*Figure 1*). While population-level descriptions of cortical circuit development are crucial, individual neurons exhibit unique developmental trajectories rooted in their specific origin, birth timing, and cellular identity (*Cossart and Garel, 2022*). In addition, the singular dynamics of sparse individual neurons can matter, as demonstrated for the rare hub cells (*Bollmann et al., 2023*; *Wang et al., 2024*). Thus, in order to fully understand the circuit basis of cortical development in health and disease, it is crucial to track neuronal activity at both population and individual cell levels in the growing brain of the same animal. This hurdle spans both experimental methodology and data analysis.

Significant progress has been made over the recent years to develop innovative solutions to meet this challenge on the experimental side, using two-photon calcium imaging. Technical improvements included modified head plates and specialised surgical and care protocols (*Bollmann et al., 2023*; *Che and De Marco García, 2021*; *Duan et al., 2020*; *Wong et al., 2018*; *He et al., 2018*; *Modol et al., 2020*). Still, in many cases, different neurons from the same mouse were recorded at different ages, or the tracking of individual cells could only be achieved through visual inspection and manual annotation

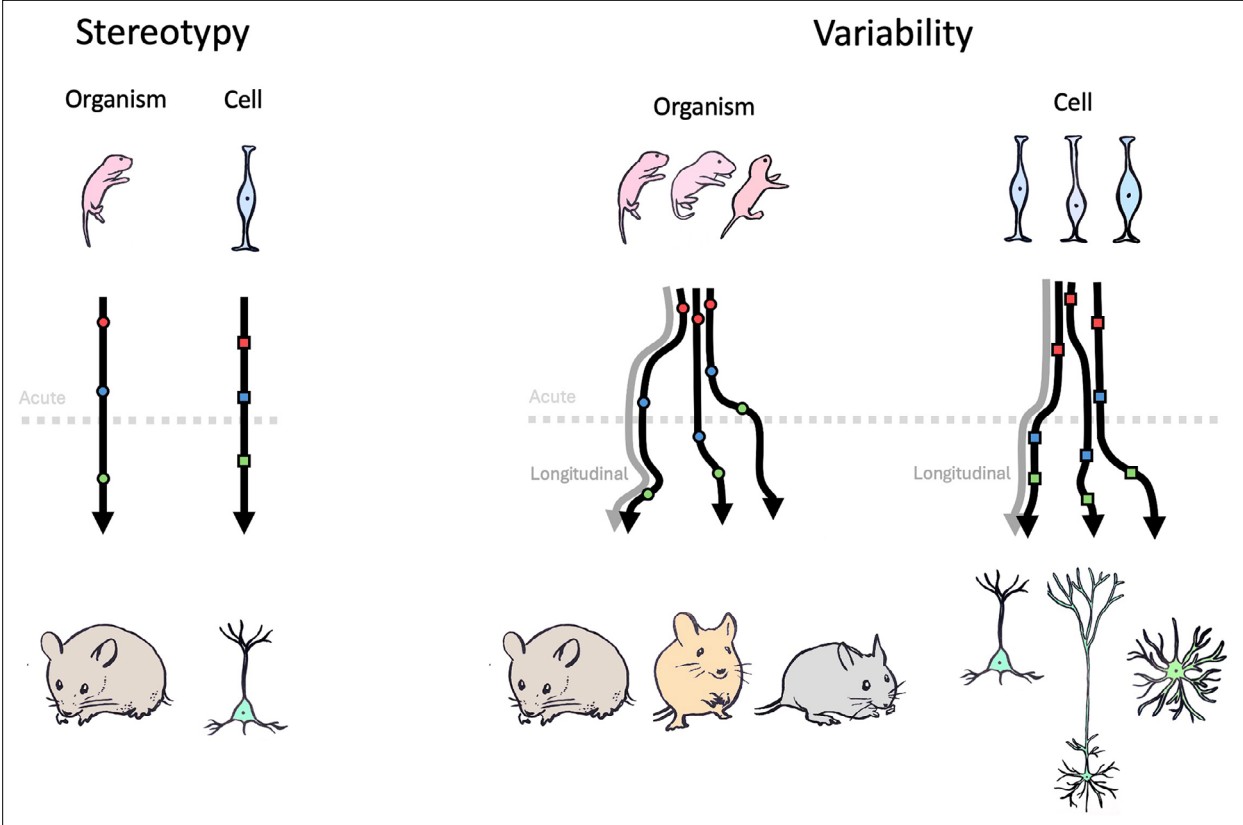

**Figure 1.** Tracking developmental trajectories at the organismal and cellular level. Postnatal development in mammals is not a strictly stereotyped process (left) but rather shows variability across individual organisms, as well as across individual cells of the same organism (right). Performing an acute experiment (dashed grey line) only provides a single snapshot of the developmental trajectory of an individual organism (or cell). Alternatively, a longitudinal experiment (grey arrow) allows tracking the properties of the same individual (or cell) throughout development, which is especially important in the case of variability in developmental trajectories (right).

of relatively sparse, often genetically subscribed neuronal populations expressing a calcium fluorescent reporter. The gold standard for developmental circuit analysis would be automated tracking of densely labelled neuronal populations, enabling efficient longitudinal monitoring while eliminating the burden of manual cell tracking. An algorithm capable of accurate automatic tracking of the same neurons across multiple days of brain development is thus essential.

Methods for tracking the activity of large populations of cells across days have been successfully developed and deployed in the adult brain (*van Beest et al., 2025*; *Lee et al., 2020*; *Ziv et al., 2013*; *Johnston et al., 2022*). In adult circuits, cell tracking is indeed relatively straightforward due to their structural stability, with minimal tissue growth, negligible morphological changes, and constant numbers of neurons. In contrast, the first weeks of mouse neocortical development are characterised by rapid and critical developmental changes spanning several scales from synapses to single neurons and networks. These include extensive brain growth (*Lyck et al., 2007*; *Thompson et al., 2014*), substantial morphological changes (*Nakazawa et al., 2018*), as well as changes to cell numbers due to programmed cell death (*Wong et al., 2018*). This makes tracking the same cells across sessions substantially more difficult during development compared to the adult brain.

To overcome the technical limitations detailed above, we developed an experimental protocol using chronic calcium imaging in mice and a novel cell tracking algorithm (Track2p), specifically tailored to development. This allowed us to track the activity of large, densely labelled, populations of neurons during early postnatal development, which has not been possible before. The algorithm overcomes the challenges of cell tracking during brain growth by applying sequential registration and cell matching steps, using the spatial overlap of cells on adjacent recordings as a matching criterion. Track2p is freely available as an open-source package with an interactive graphical user interface

(GUI), enabling researchers to analyse longitudinal calcium imaging data from both developing and mature circuits.

Applying the algorithm to a dataset recorded during the second postnatal week in mouse barrel cortex, a critical period for the formation of topographic maps in that region, yielded hundreds of identified neurons tracked across all days. Assessing the quality of the algorithm by benchmarking it on a newly generated ground truth dataset showed high tracking performance (*Sheintuch et al., 2017*). Leveraging ground truth benchmarking, we demonstrate that explicitly accounting for developmental processes, such as brain growth, is critical for accurately tracking cells during postnatal development. Our work thus shows that chronic calcium imaging and cell tracking using Track2p can be used to monitor the changing physiological properties of large populations of matched neurons during early cortical development. We demonstrate that the statistics of activity patterns in the tracked population display two periods of stability, with a critical transition point around postnatal day 11 (P11), marking the emergence of a stable behavioural state representation.

## Results

### Cell tracking using image registration and overlap-based matching

Tracking neuronal activity across multiple days presents unique challenges due to the dynamic nature of brain development. We developed a novel tracking algorithm, called Track2p. As in other tracking algorithms for calcium imaging data (*van Beest et al., 2025*; *Johnston et al., 2022*; *Tolu et al., 2010*), the final goal of Track2p is to follow individual cells across sessions allowing the user to compare their functional properties in downstream analyses. To achieve this goal, the algorithm takes as input a set of preprocessed recordings, each consisting of a set of regions of interest (ROIs, i.e. putative neurons detected based on activity, see Methods) and their respective calcium fluorescence traces, as well as a mean image of the field of view (FOV). Briefly, the algorithm aims to match ROIs in any given pair of consecutive sessions based on their spatial overlap, assuming that the more the two overlap in anatomical space, the more likely they correspond to the same neuron. Due to developmental processes such as brain growth and other experimental factors, it is necessary to account for day-to-day changes that occur between the two recordings before computing spatial overlaps. This is achieved by performing affine image registration on the mean FOV images between consecutive days.

We apply the registration and spatial matching iteratively, starting with the first pair of sessions ($s_0$ and $s_1$) as follows (*Figure 2A*).

Firstly, we estimate the spatial transformation between $s_0$ and $s_1$ using affine image registration (*Ntatsis et al., 2023*) (i.e. allowing shifting, rotation, scaling, and shearing, see *Figure 2B*, the transformation is denoted as T). We employ affine transformation, since it can account for both rigid transformations (rotations and translations arising from minor mismatches in FOV alignment across experiments) and scaling and shearing (mostly due to brain growth). The changes across the two consecutive recordings are approximated as the transformation registering the mean FOV from session $s_1$ (green in *Figure 2B*) to the mean FOV of $s_0$ serving as reference (red in *Figure 2B*).

Secondly, the computed transformation is applied to the ROIs from session $s_1$ (green in *Figure 2C*) to align them to ROIs from session $s_0$ (red in *Figure 2C*). The amount of spatial overlap after registration (yellow in *Figure 2C*, bottom) can indicate the accuracy of the estimated transformation between the 2 days. Assuming that the transformation is accurately estimated, ROIs corresponding to the same cells display substantial spatial overlap, with some ROIs from one recording also potentially overlapping poorly, if not at all, with any ROI from another.

Thirdly, once the ROIs are aligned, the algorithm proceeds with the matching (*Figure 2D*). This is done by computing a spatial similarity metric (intersection over union [IoU]) between each ROI from session $s_0$ and each transformed ROI from session $s_1$. Matches are then assigned in a globally optimal way by maximising the sum of IoU values across all matches using a linear sum assignment algorithm (*Crouse, 2016*).

Finally, since two consecutive sessions contain different sets of detected cells (see * in *Figure 2D*, bottom) and since ROIs can overlap even if the signal does not come from the same cells (see + in *Figure 2D*, bottom), we perform an additional filtering step on the assigned matches. Assuming that the IoU values for putative true and putative false matches come from different distributions, we would expect a bimodal distribution of IoU values across all assigned matches (see histogram in

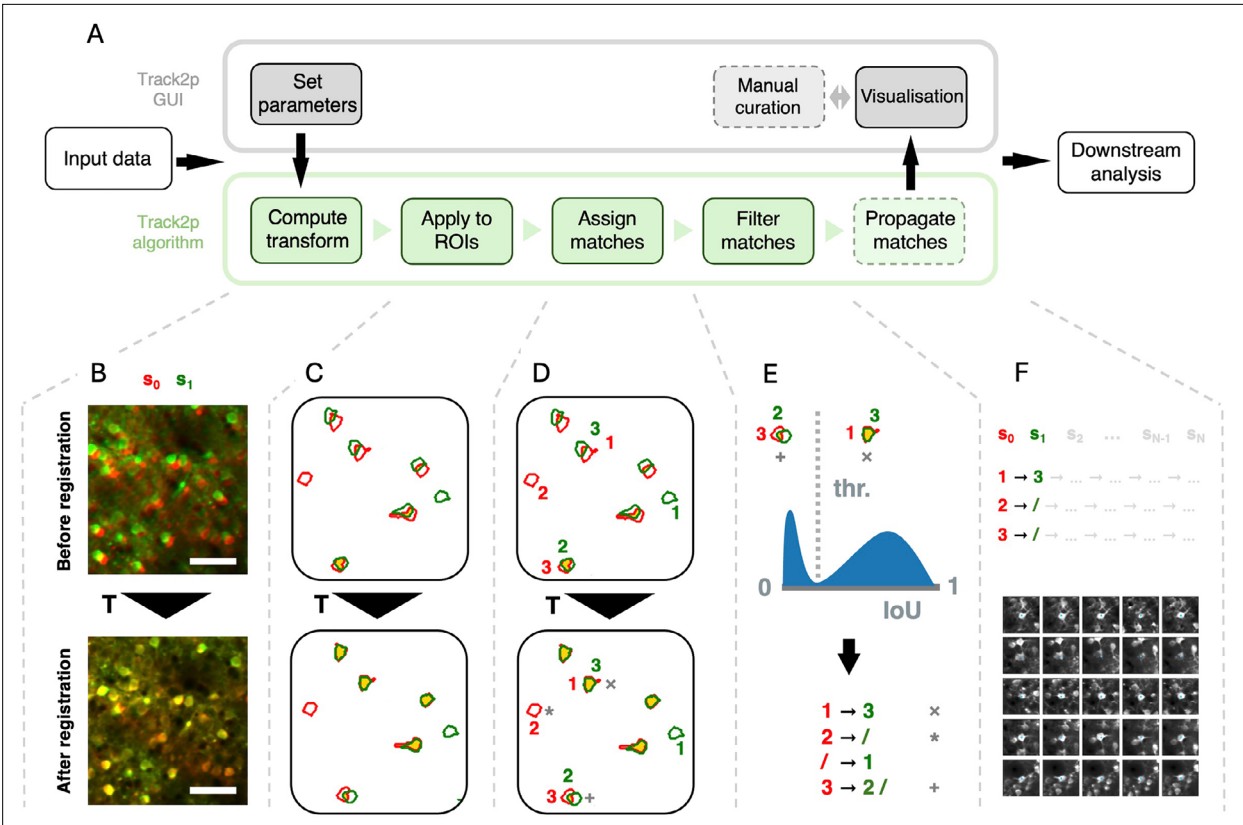

**Figure 2.** The Track2p pipeline for tracking cells across recordings. (**A**) Schematic overview of the Track2p algorithm and the graphical user interface (GUI) capabilities. Dashed squares represent optional steps in the pipeline. (**B**) The transformation (**T**) between 2 consecutive days is computed by registering the mean field of view (FOV) images. Top: An overlay of the reference (red, $s_0$) and the moving (green, $s_1$) images before registration. Bottom: Same two images after registration. Scale bar: 50 μm. (**C**) Applying the transform to regions of interest (ROIs) from Suite2p segmentation. The ROI colour code is the same as in B, the intersection of the ROIs is shown in yellow. Top: Overlap before T. Bottom: Overlap after T. Note: Only a few example ROIs are displayed for explanation purposes. (**D**) Cell matching using linear sum assignment. ROIs are the same as C with indexes for the two recordings added for illustrative purposes. Cells from one recording are matched to cells from another, maximising the sum of the intersection over union (IoU) across all matches. (**E**) Filtering putative false and true matches by thresholding the IoU distribution. Top: The distribution of IoU values for matched ROI pairs shows a bimodal distribution, which is used to reject putative false positive matches. Bottom: Final result of the cell linking for a pair of recordings. (**F**) Top: In the case of tracking across more than two imaging sessions, the steps from B to D are repeated sequentially to link the cells across all days. Bottom: Example matches for five cells (rows) successfully tracked across 5 consecutive days (columns). Note: The figures shown are for illustrative purposes only; see *Figure 3* for application to real data.

The online version of this article includes the following figure supplement(s) for figure 2:

**Figure supplement 1.** Overview of the graphical user interface for interactive visualisation and curation of tracked cells.

*Figure 2E*). To reject the putative false positives, we compute a threshold based on automatic thresholding methods (Otsu's method; *Ulman et al., 2017*). This ensures that assigned matches with low spatial similarity are rejected (see + in *Figure 2D and E*) while the matches with high similarity are accepted (see x in *Figure 2D and E*). This yields the final matching for the first pair of consecutive imaging sessions ($s_0$ and $s_1$). In the case of more than two recordings, this tracking procedure is then iteratively applied for all consecutive pairs of sessions ($s_0$ to $s_1$, $s_1$ to $s_2$ ... $s_{N-1}$ to $s_N$, and so on), with tracks being extended sequentially from $s_0$ to $s_N$ based on the identified matches, and terminated if a match was not identified at any particular session (*Figure 2F*).

We provide an open-source implementation of the algorithm, combined with a user-friendly GUI, allowing non-specialist users to run the algorithm and interact with its outputs (see *Figure 2A* top, *Figure 2—figure supplement 1*, and Appendix 1). Both the algorithm and the GUI come with a simple installation procedure and extensive documentation to facilitate the ease of use and accessibility of Track2p.

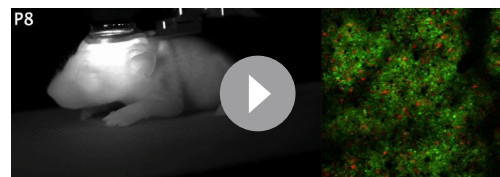

**Video 1.** Example of videography and two-photon calcium imaging data acquired throughout the second postnatal week. Note the substantial growth of the mouse, the development of the whisker pad, and opening of the eyes, and the loss of synchrony and spatial organisation in the calcium imaging data. For visualisation purposes, the videos were manually aligned using Fiji.

https://elifesciences.org/articles/107540/figures#video1

## Tracking neurons across days in the early postnatal neocortex

In order to validate our method on experimental data, we next applied Track2p to a longitudinal dataset consisting of daily recordings of the same 720×720 μm² FOV throughout the second postnatal week of mouse barrel cortex development (P8 to P14, n=7 imaging sessions from one mouse; for more details, see Methods and (*Video 1*)) In addition to the calcium indicator (GCaMP8m), we virally expressed a sparse anatomical marker (tdTomato) targeting GABAergic neurons using conditional expression in GAD67-Cre mice (*Pachitariu et al., 2017*). This dual-labelling strategy facilitated reliable FOV identification across sessions and provided an anatomical reference for Track2p's initial cross-session registration and tracking (*Figure 3*, for comparison with tracking based on GCaMP8m registration, see the 'Benchmarking on manually tracked cells' section). Before running Track2p, each recording was preprocessed using Suite2p (*Otsu, 1979*) for active cell detection (segmentation) and calcium fluorescence trace extraction (see Methods). The outputs of Suite2p (ROIs, traces, and FOV images) were then used as inputs to the Track2p algorithm.

The Track2p algorithm appeared to successfully register the mean FOV for each successive pair of imaging sessions (based on visual inspection of Track2p outputs; for the first and last pairs, see *Figure 3A and B*, bottom left; for the equivalent of *Figure 3* across all days, see *Figure 3—figure supplement 1A*). Registering the ROIs for each pair of consecutive recordings showed a great degree of overlap for the majority of ROIs (yellow area in *Figure 3A and B*, bottom). We did, however, observe several ROIs that were only present in one session from the pair, likely due to differences in cell detection and developmental factors such as growth, silencing, or cell death. When matching ROIs across sessions using spatial overlap (IoU), we would expect these ROIs to show significantly lower IoU values compared to the cells that were present in both sessions. Indeed, the IoU distribution of assigned matches revealed a bimodal distribution for each pair of imaging sessions, allowing us to use classical histogram thresholding methods (*Ulman et al., 2017*) to clearly separate the distributions of putative true and false matches (*Figure 3A and B*, bottom right, *Figure 3—figure supplement 1B*). Propagating the putative matches yielded a total of 728 ROIs that were tracked across all days in this example mouse (out of 1988 ROIs detected in the first recording, *Figure 3E*). We used the subset of cells successfully tracked across all days for all our future analyses.

Plotting ROIs overlaid on top of the mean image of the GCaMP8m imaging channel (*Figure 3D* for contours on whole FOV, C for magnified example matches) and visually inspecting the matches using the Track2p GUI indicated excellent tracking with Track2p. These visualisations also revealed the substantial growth of the neocortex during the course of the experiment (FOV area covered by ROIs at P8 compared to P14 in *Figure 3D*), with some cells growing out of the FOV (see top and left in *Figure 3D*) and the pairwise distances between matched ROIs increasing by approximately 15% (*Figure 3F*).

Applying Track2p on an example longitudinal imaging dataset thus demonstrated that it can successfully track activity from a large number of putatively matched neurons throughout early postnatal development in mice, as confirmed by visual inspection. To assess the tracking performance in a more quantitative and objective way, we next benchmarked Track2p's performance on manually generated ground truth for our specific experimental setting.

## Benchmarking on manually tracked cells

In order to benchmark Track2p, expert annotators manually tracked cells across sessions (similarly as in the Cell Tracking Challenge; *Sheintuch et al., 2017*; *Rochefort et al., 2009*). Benchmarking was performed based on a dataset from three mouse pups imaged under the same experimental conditions as described in the previous section (including the dataset shown in *Figure 3*). For each

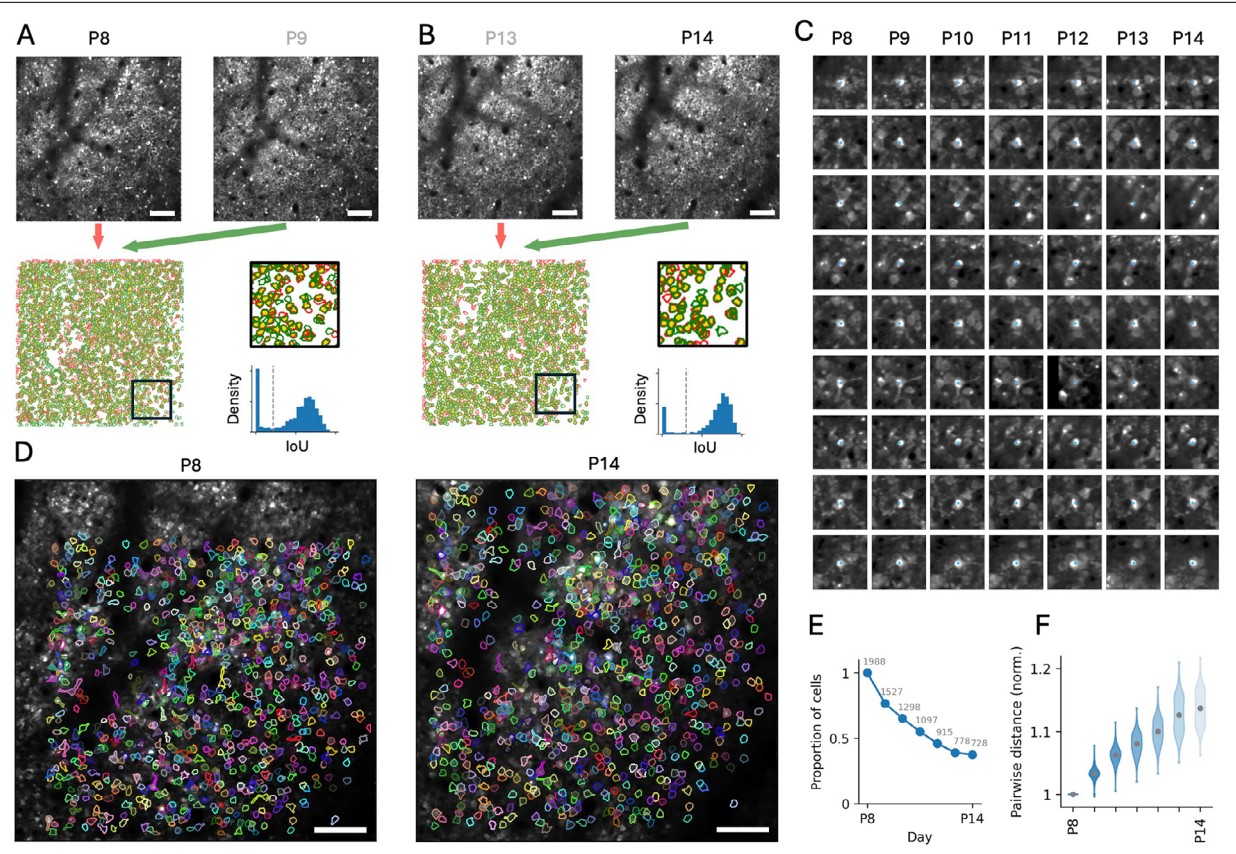

**Figure 3.** Track2p tracks hundreds of cells throughout the second postnatal week of development in the mouse neocortex despite substantial brain growth. (**A**) Top left and right: Mean images of the 'anatomical' channel (tdTomato in GABAergic cells) for the first two imaging days (postnatal day 8 [P8] and P9). Scale bar: 100 μm. Bottom left: Overlay of the two images after registration (pseudocoloured as red and green, respectively). Bottom middle: Overlay of the regions of interest (ROIs) after registration (same colour code for P8 and P9). Bottom right: Distribution of intersection over union (IoU) values for matched pairs showing the automatic threshold (grey dashed line). (**B**) Same as A but for the last two imaging days (P13 and P14). (**C**) Nine representative examples of matches visualised in the mean 'functional' channel (signal from pan-neuronal GCaMP8m expression) on the first and last days of recording (P8 and P14). The blue dot indicates the centroid of the ROI. (**D**) Overlay of all ROIs successfully tracked across all days (N=728) on the mean 'functional' channel image of the first (P8, left) and last (P14, right) imaging days. Each tracked ROI is shown in the same colour across the plots. Note the expansion of the field of view (FOV) at P14 compared to P8. Scale bar: 100 μm. (**E**) Graph plotting the proportion (y-axis) and absolute number (grey text) of cells successfully tracked from the first day of imaging onwards. (**F**) Brain growth quantified as the relative increase in pairwise distances between tracked cells normalised to the first day. Grey dots represent the mean for each recording. Note: For visualisations in panels A, B, C, and D across all days, see *Figure 3—figure supplement 1*.

The online version of this article includes the following figure supplement(s) for figure 3:

**Figure supplement 1.** Track2p outputs across all days for the example mouse.

experiment, we chose 64 homogeneously distributed ROIs detected on the first day and tracked them based on visual inspection across consecutive sessions (for more details, see Methods and *Figure 4—figure supplement 1*). This left us with, on average, 20 neurons per experiment that we were able to manually track across all days ('GT' in *Figure 4*). We then proceeded to compare these to cell tracks identified by Track2p. For the purposes of evaluation, we used a fully automatic tracking procedure (without manual curation and with default Track2p parameters).

Different metrics exist for evaluating cell tracking. Since we are aiming to track cells across all days, a robust cell tracking metric should reward perfect track matches with ground truth while imposing penalties for missed or incorrectly identified tracks. For this reason, we used a biologically inspired 'complete tracks' quality metric, which corresponds to the F1 score for completely reconstructed full tracks (we refer to this value as 'CT' according to *Sheintuch et al., 2017*; *Rochefort et al., 2009*). A CT score of 1 would correspond to perfect tracking, while a CT score of 0 would correspond to no matches or a large proportion of wrong matches (see *Figure 4A* and Methods for more details).

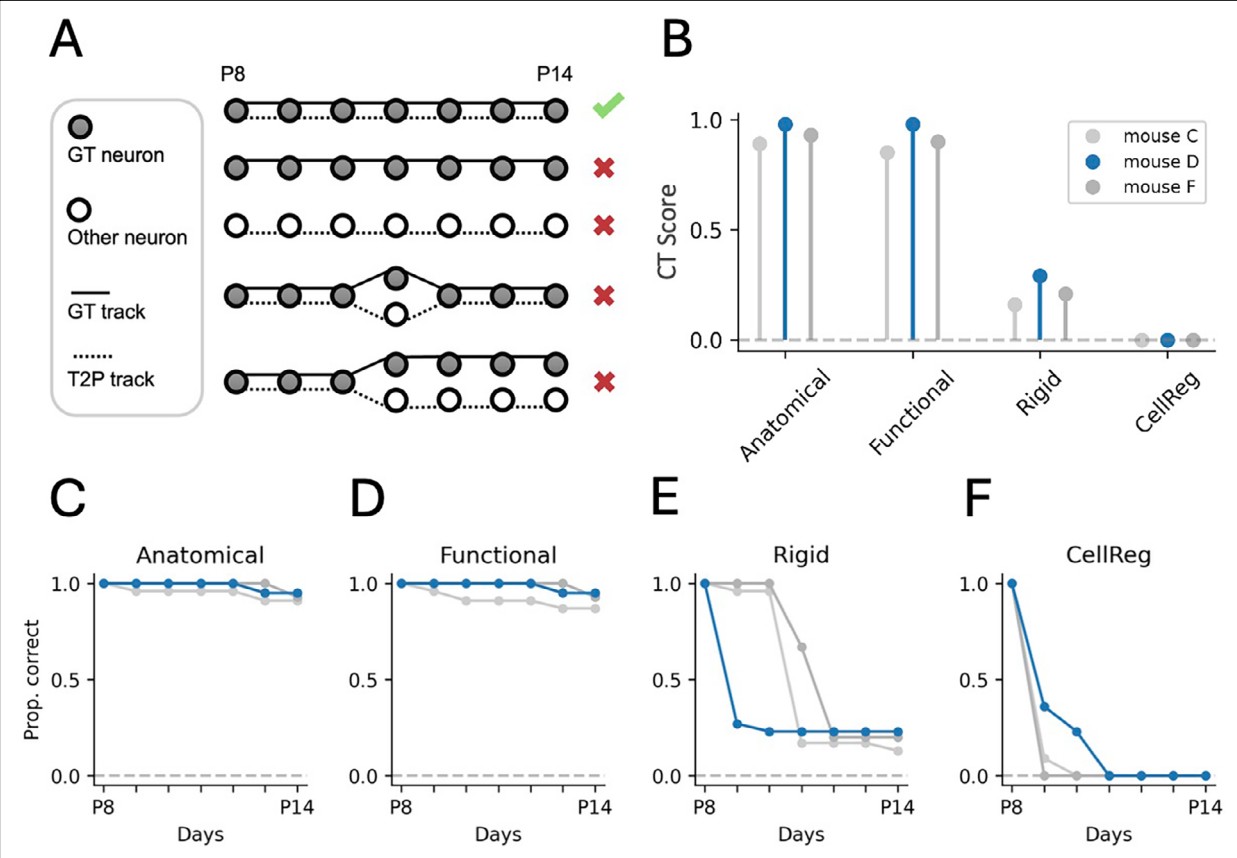

**Figure 4.** Evaluation of tracking performance on a manually tracked ground truth dataset during the second postnatal week of development in the mouse neocortex. (**A**) A schematic representation of possible cases when comparing ground truth tracks (GT, solid lines) with those reconstructed by Track2p (T2P, dashed lines). The CT metric favours perfect matches (top row) and penalises all types of mismatches (bottom four rows). (**B**) Graph showing the CT score for all three GT datasets across evaluated conditions. Mean CT scores of 0.93, 0.91, 0.22, and 0.00 for 'Anatomical', 'Functional', 'Rigid', and 'CellReg', respectively. Blue denotes the example mouse illustrated in *Figure 3*. (**C**) Proportion of fully correctly reconstructed GT traces for increasing time spans starting from postnatal day 8 (P8) for the baseline ('Anatomical') condition. (**D, E, and F**) Same as C but for the 'Functional', 'Rigid', and 'CellReg' conditions, respectively.

The online version of this article includes the following figure supplement(s) for figure 4:

**Figure supplement 1.** Generating a ground truth dataset for Track2p evaluation.

Additionally, to assess tracking performance over time, we quantified the proportion of reconstructed ground truth tracks over progressively longer time intervals (first 2 days, first 3 days, etc. 'Prop. correct' in *Figure 4C–F*, see Methods). This allowed us to understand how tracking accuracy depends on the number of successive sessions, as well as at which time points the algorithm might fail to successfully track cells.

Track2p was evaluated under three different scenarios and compared to the performance of a widely used algorithm developed for longitudinal tracking in the adult brain (CellReg; *Tolu et al., 2010*, see Methods). The initial evaluation was done using day-to-day registration based on a sparse anatomical marker (as in *Figure 3*). Calculating the CT score in this condition showed remarkably high tracking performance for all datasets (see 'Anatomical' in *Figure 4B*, mean CT = 0.93). To test whether a sparse marker was strictly necessary for successful tracking, we next ran the algorithm using the mean image of the GCaMP8m channel as a comparison. Interestingly, this evaluation showed a similar performance (see 'Functional' in *Figure 4B*, mean CT = 0.91), indicating that dense calcium indicator labelling alone can provide comparable tracking performance, eliminating the requirement for sparse anatomical labelling.

To show the importance of a method tailored to the growing brain, we next compared the baseline tracking performance with two alternative conditions: firstly, by using Track2p without explicitly

accounting for day-to-day growth, performing a rigid instead of affine registration of consecutive recordings; and secondly, by comparing Track2p to CellReg tracking (*Tolu et al., 2010*). In both scenarios, tracking performance significantly deteriorated compared to our baseline method (see 'Rigid' and 'CellReg' in *Figure 4B*, mean CT = 0.22 and CT = 0, respectively). Interestingly, however, these methods still managed to correctly reconstruct a portion of tracks across shorter age spans in certain cases (*Figure 4E and F*), with the performance dropping significantly for longer tracks in comparison to the baseline condition (*Figure 4C and D*).

These comparisons demonstrate Track2p's robust cell tracking performance in the developing brain, maintaining high accuracy over extended periods of development. However, tracking performance certainly depends on the type of experimental data (age, model system, brain area, imaging parameters, FOV alignment, etc.), hence, we also provide additional resources and documentation (https://track2p.github.io/home.html), allowing users to benchmark Track2p tracking for their particular use case.

## Development of firing statistics from hundreds of tracked neurons across postnatal development

Having validated the performance of cell tracking, we next analysed the development of functional properties of the tracked population of neurons across the second postnatal week of mouse barrel cortex development. For this, we used a full dataset of six mice imaged daily for a minimum of 6 consecutive days within the second postnatal week (P7 to P14, see *Figure 5B* for summary; we used this dataset for all subsequent analyses). On average, 526 (±190 std) neurons per mouse were successfully tracked across all days using Track2p, corresponding to 33% (±11% std) of the neurons detected on the first day for each mouse. Based on the high tracking performance on ground truth (see *Figure 4*), we conclude that this drop is due to a failure in detection rather than tracking. During the course of the experiment, the weight of a mouse increased on average by 55% (±18% std), with the pairwise distance between neurons increasing on average by 15% (±7% std) (*Figure 5C*, *Figure 5—figure supplement 2F*). Weight was found to vary significantly across mice, as early as the first experimental day, and to correlate with brain growth (r=0.92, *Figure 5—figure supplement 2F*). Our robust tracking capability provides a comprehensive view of the evolution of neuronal dynamics within single mice, while also allowing us to take into account the heterogeneity of the developmental timelines across mice.

The early postnatal period of cortical development studied here is characterised by the transition from synchronous calcium events recruiting many neurons to progressively more decorrelated population dynamics (*Golshani et al., 2009*; *Mizuseki and Buzsáki, 2013*). In order to visualise these global changes in our dataset of longitudinally imaged mice, we first plotted raster plots of calcium fluorescence traces as a function of time for all tracked neurons on each imaging day in our example mouse (*Figure 5A* for P8 and P14, all days in *Figure 5—figure supplement 1A*). Visual inspection of these raster plots clearly indicates the disappearance of recurring periods of highly synchronous activity from around P11 onwards, as well as a global increase in single-neuron activity rates. Accordingly, quantifying calcium event rates as a function of age indicated a gradual increase in the mean and a widening of the distribution with age (*Figure 5D*, *Figure 5—figure supplement 2A and G*, all statistical comparisons were done between an 'early' [≤P11] and 'late' [>P11] set of recordings). This evolution also signalled the transition towards long-tailed firing rate distributions, which are ubiquitous in adult brain circuits (*Avitan and Stringer, 2022*).

Next, we analysed the distributions of pairwise correlations between activity traces from all longitudinally tracked neurons as a function of mouse age (*Figure 5D*, *Figure 5—figure supplement 2B and H*). Consistent with the disappearance of highly synchronous network events observed in the raster plots, we found a significant decrease in mean pairwise correlations, indicating a progressive decorrelation of neuronal activity (*Figure 5E*). The spatial distribution of pairwise correlations also evolved across days from highly correlated locally to more broadly distributed (*Figure 5F*), suggesting the gradual breakdown of spatially clustered cell assemblies (*Wu et al., 2024*).

We finally turned to characterising the dominant population patterns of neural activity using principal component analysis (PCA). Interestingly, we observed an increase in the number of components required to explain a fixed amount of variance in the neural data across days, suggesting a developmental increase in dimensionality (*Figure 5G*), consistent with the statistics of spontaneous

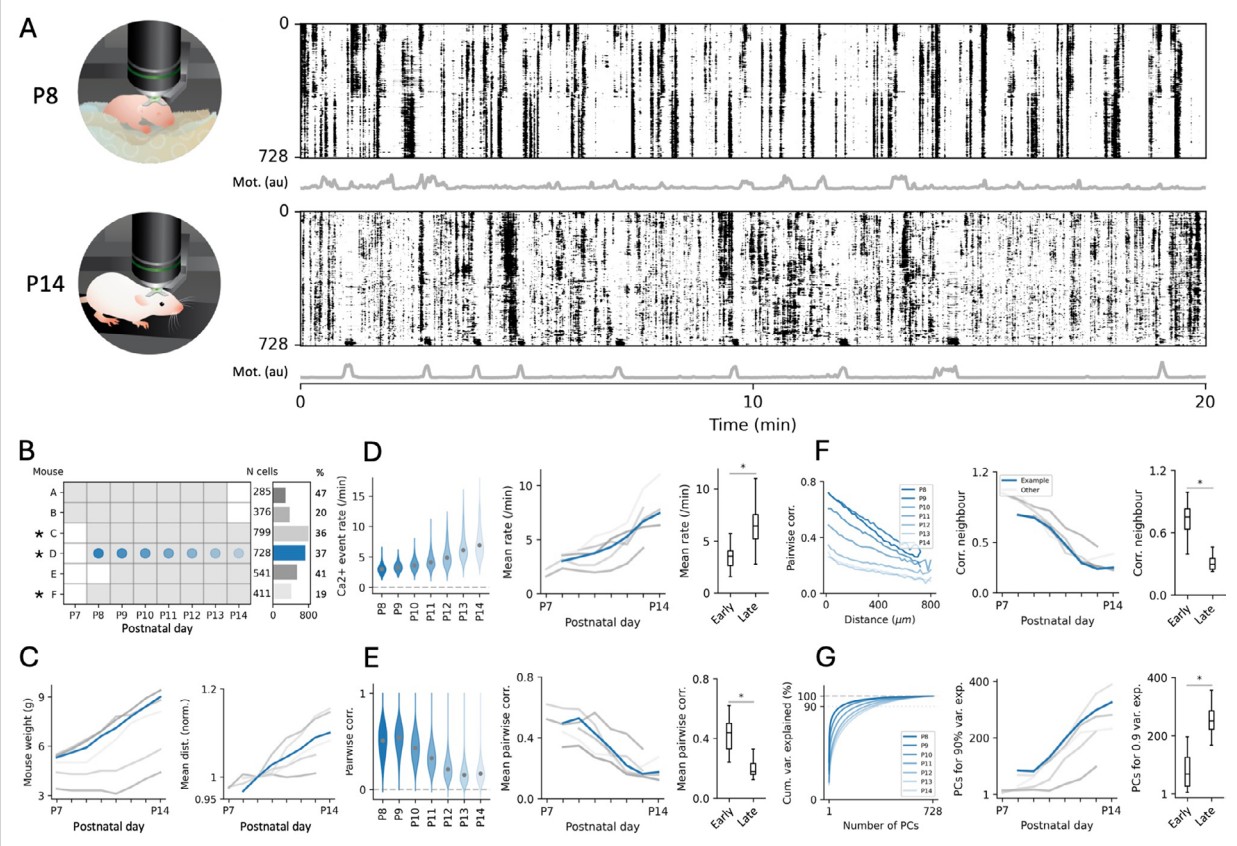

**Figure 5.** Evolution of neuronal activity statistics from hundreds of tracked neurons during the second postnatal week of mouse development. (**A**) Raster plots showing the activity of all 728 tracked neurons as a function of time for the example mouse at postnatal day 8 (P8) (top) and P14 (bottom). Each row in the raster corresponds to the trace of a single cell with the sorting determined by their Rastermap embedding computed at P14. Grey traces underneath the raster show a metric of global motion of the mouse computed from videography (see Methods). (**B**) Overview of the dataset, indicating the imaging days for each mouse (left), the total number of cells successfully tracked across all recording days (N cells, right); * indicates mice used in the evaluation of the algorithm (*Figure 3*); blue denotes the example mouse. % cells: N cells as a proportion of all cells detected on the first day. (**C**) Graphs plotting mouse weight (left) and the mean pairwise distance between tracked neurons (right; normalised to P9 corresponding to the earliest common day across all mice) as a function of imaging days t. (**D**) Graphs plotting the distribution of calcium fluorescence event rates in all tracked neurons from the example mouse as a function of age (left), the mean value across days for all mice (middle), and a statistical comparison between the early (≤P11) and late (>P11) epochs (right). Example mouse is shown in blue; same in E, F, and G. *: Mann-Whitney U test, $p=7.6 \times 10^{-6}$. For standard deviation, see *Figure 5—figure supplement 1G*. (**E**) Same as D but for pairwise correlations. *: Mann-Whitney U test, $p=1.8 \times 10^{-6}$. For standard deviation, see *Figure 5—figure supplement 1H*. (**F**) Graphs plotting pairwise correlations as a function of anatomical distance for all pairs of tracked neurons across all ages in the example mouse (left), the estimated pairwise correlation of neighbouring neurons as a function of age for all mice (middle), and a statistical comparison across the two epochs (right). *: Mann-Whitney U test, $p=2.7 \times 10^{-7}$. (**G**) Cumulative distribution plot of the explained variance as a function of the number of principal components (PCs) for the example mouse across ages (left), number of PCs accounting for 90% of the variance as a function of age for all mice (right), and a statistical comparison across the two epochs (right). *: Mann-Whitney U test, $p=4.0 \times 10^{-6}$.

The online version of this article includes the following figure supplement(s) for figure 5:

**Figure supplement 1.** Activity of tracked cells across all days for the example mouse.

**Figure supplement 2.** Development of neuronal activity statistics in the tracked population of neurons for the full dataset.

activity described in the adult brain (*Stringer et al., 2019a*). Notably, the plots of summary statistics for our entire dataset indicated a clear outlier, consistent across all quantifications (*Figure 5D, E, F, and G*). This outlier mouse displayed a similar but delayed developmental trajectory, compared to the other mice. Interestingly, it also showed the lowest initial weight and a less pronounced growth in weight and cortical size (*Figure 5C*), likely indicating a lower maturation stage at the onset of the experiments, although the contribution of other experimental factors cannot be excluded given the invasiveness of imaging surgery.

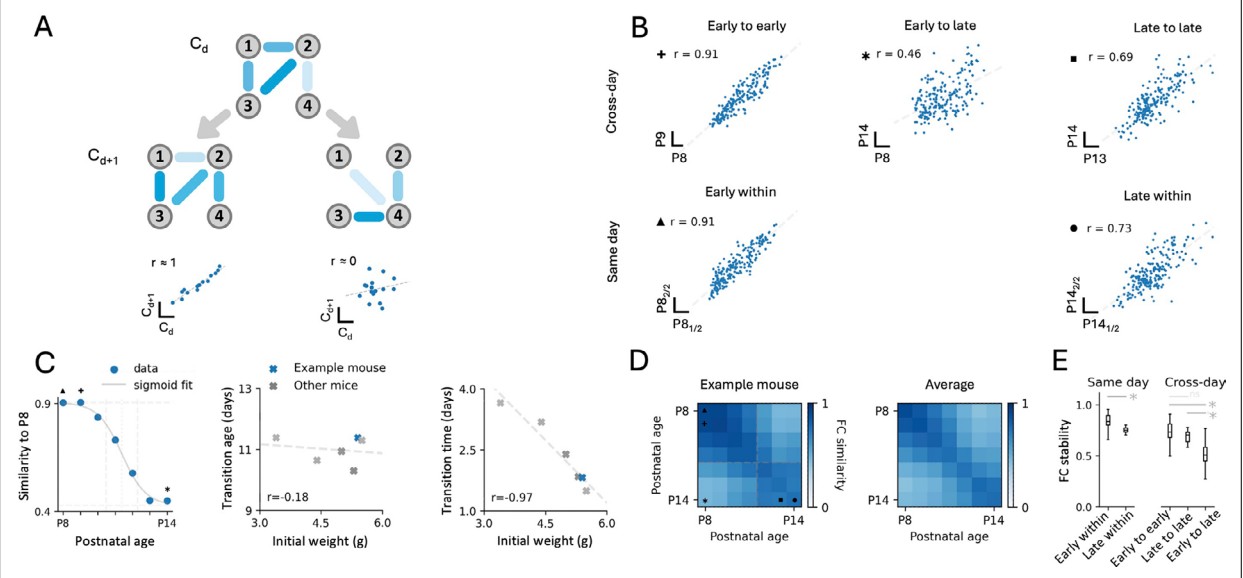

**Figure 6.** Transition between two stable functional network structures during the second postnatal week of barrel cortex development. (**A**) Schematic explanation of the framework to compare functional connectivity (FC) across days; grey nodes represent neurons, strength of a functional connection is denoted by the opacity of the blue edge between two nodes. For a network of four neurons and a given connectivity $C_d$ (16 connections, top), we can imagine that the connectivity on the next day ($C_{d+1}$) could be 'conserved' (middle left) or it could be 'reorganised' (middle right). A scatter plot comparing all 16 functional connections between $C_d$ and $C_{d+1}$ would indicate a high correlation in the 'conserved' case (bottom left) and low correlation in the reorganised case (bottom right). We refer to this correlation as 'FC similarity'. (**B**) Scatter plots of FC for three different pairs of recording days: a pair of early sessions (postnatal day 8 [P8] to P9, top left), an early and a late session (P8 to P14, top middle), and a pair of late sessions (P13 to P14, top right). Within-session FC similarity (bottom scatter plots), i.e., comparing the first and second half of an imaging session at early ($P8_{1/2}$ to $P8_{2/2}$, bottom left) and late ages ($P14_{1/2}$ to $P14_{2/2}$, bottom right). For visualisation purposes, a random subset of 200 pairs is displayed; Linear fit and r (Pearson correlation or 'FC similarity') are computed on all pairs; symbols next to r values indicate the same values in panels C and D. (**C**) FC similarity between P8 and all other days for the example mouse (left). Sigmoid fit: solid grey line; inflection point ('transition age', see Methods): dotted grey vertical line; linear portion of the sigmoid ('transition time', see Methods: two dashed grey vertical lines). Transition age (middle) and transition time (right) as a function of initial weight at P7 are plotted for all mice. (**D**) FC similarity matrix (left) for all pairs of recording days (off diagonal), with within-session FC similarity on the diagonal for the example mouse (left) and average FC similarity across all mice for the same period (right). (**E**) Box plots comparing within-day (left) or across-day (right) FC similarity for early (≤P11) and late (>P11) developmental epochs pooled across all mice. *: statistically significant; ns: not statistically significant (Kruskal-Wallis test: $p=3.6 \times 10^{-22}$; post hoc Mann-Whitney U tests with Bonferroni correction, 'early within' to 'late within': $p=3.3 \times 10^{-3}$, 'early to early' to 'late to late': p=n. s., 'early to early' to 'early to late': $p=1.7 \times 10^{-12}$, 'late to late' to 'early to late': $p=4.2 \times 10^{-5}$, for all possible comparisons, see *Figure 6—figure supplement 1K*).

The online version of this article includes the following figure supplement(s) for figure 6:

**Figure supplement 1.** Stability of neural activity statistics for all mice in the dataset.

To fully leverage our longitudinal approach, we next turned to comparing the functional properties of the same neurons across development, which is only possible when tracking cells across days.

## A marked reorganisation of the functional network structure during the second postnatal week

We first examined the stability of the correlation structure, which we will refer to as 'functional connectivity' (FC). Two alternative possibilities could be envisaged (see *Figure 6A*): (i) FC could be conserved across days, meaning that a highly connected pair on a given day would remain highly connected on the next (*Figure 6A*, left); (ii) FC could reorganise, losing the structure from the previous day (*Figure 6A*, right). In the first case, the FC values for all pairs would remain similar across the 2 days (*Figure 6A*, bottom left), whereas in the second case, they would be different (*Figure 6A*, bottom right). To discriminate between the two cases, we defined an 'FC similarity' score for any pair of imaging days with matched neurons for a given mouse (Pearson correlation (r) across all neuron pairs, see *Figure 6A*, bottom). Such an FC similarity metric encapsulates a variety of possible sources of network changes, from local or long-range connectivity, intrinsic excitability to neuromodulation.

Quantifying FC similarity across two early sessions for an example animal (P8 to P9, *Figure 6B*, top left) indicated remarkable stability, almost identical to the FC similarity within a given imaging day session which we took as a reference (computed as FC similarity between the first and second half of a same day recording, *Figure 6B*, bottom left). Conversely, when comparing across more distant developmental ages, we noticed that a large part of the correlation structure was lost, resulting in lower FC similarity (P8 to P14, *Figure 6B*, top middle). Interestingly, FC similarity between a pair of later developmental sessions was again comparable to within-session similarity (P13 to P14 and P14 within, *Figure 6B*, right), indicating that the functional network structure was stable, but different from that of earlier ages. Of note, within-session stability was consistently lower later than earlier. Quantifying calcium event rate similarity across days indicated a similar pattern (see *Figure 6—figure supplement 1F–J*).

To investigate this further, we proceeded to compute the FC similarity for all possible combinations of sessions in all mice. Interestingly, we observed a sigmoid-like decay of FC similarity when taking P8 as a reference (*Figure 6C*). Plotting the full FC similarity matrix for all possible combinations of sessions also indicated two stable FC regimes with seemingly two blocks along the diagonal (*Figure 6D*, left for example, *Figure 6D* right for average, and *Figure 6E* for statistical comparison). Interestingly, the sharpness of the transition in FC varied across mice (*Figure 6—figure supplement 1C*) and could in part be explained by the weight of the mouse at the onset of the experiment (*Figure 6C*, linear part of sigmoid fit). Such variation across mice could reflect either inherent individual differences in developmental processes or experimental factors, where mice with a lower initial weight may have experienced delayed development due to higher sensitivity to the imaging procedure.

Altogether, our analyses indicate that the second postnatal week marks a transition between two stable functional network structures in the barrel cortex of developing mice. Additionally, this shows how tracking cells during development opens new analysis possibilities using self-referencing between neurons, providing new insights into how the developmental choreography unfolds in the same population of neurons across days.

## Developmental emergence of stable behavioural state modulation

As a first attempt to investigate the mechanisms underlying this network transition, we examined neural activity in relation to behavioural state. The second postnatal week marks the emergence of active sensation, suggesting that changes in arousal state modulation might drive this network reorganisation. In the adult cortex, arousal strongly shapes both global firing rates and neural correlations (*Benisty et al., 2024*; *Niell and Stryker, 2010*; *Gentet et al., 2010*; *Stringer et al., 2025*), making it a promising candidate for orchestrating the developmental shift we observed.

To this aim, we sorted the neurons based on the similarity of their activity patterns, using Rastermap (*Lütcke et al., 2013*), and examined the relationship between neuronal activity and behavioural states. We assessed behavioural state indirectly using the videos capturing spontaneous mouse movement and quantifying them using a 'motion energy' metric (as in *Benisty et al., 2024*, for more details, see Methods). Interestingly, this sorting revealed a subpopulation of neurons that were highly correlated with the animal's motion at later, but not earlier, developmental stages (*Figure 5A*, see *Figure 5—figure supplement 1A* for all days). We first quantified this relationship by computing the temporal correlation between the first principal component of neuronal activity and the animal's motion. Interestingly, this correlation showed a steep increase after P11 for most mice (*Figure 7D*). This suggested that global fluctuations in neural activity are more strongly modulated by active movement at later than earlier developmental stages.

Finally, we asked whether behavioural state could be decoded from population activity dynamics using regression analysis. Despite substantial variability, the same-day decoding performance increased with development (*Figure 7A–C*). Since we tracked the same neurons across days, we could also probe the stability of this representation across days, by fitting a model on 1 day and testing it on all other days (cross-day decoding, *Figure 7E–H*, *Figure 7—figure supplement 1B and C*; *Ziv et al., 2013*; *Lütcke et al., 2013*). Interestingly, this analysis showed that, once developed, this representation was indeed stable, allowing for accurate cross-day decoding, with the same neurons showing either consistently positive or negative modulation across days (see *Figure 7—figure supplement 1D* for the activity traces of an example cell with a high weight for decoding).

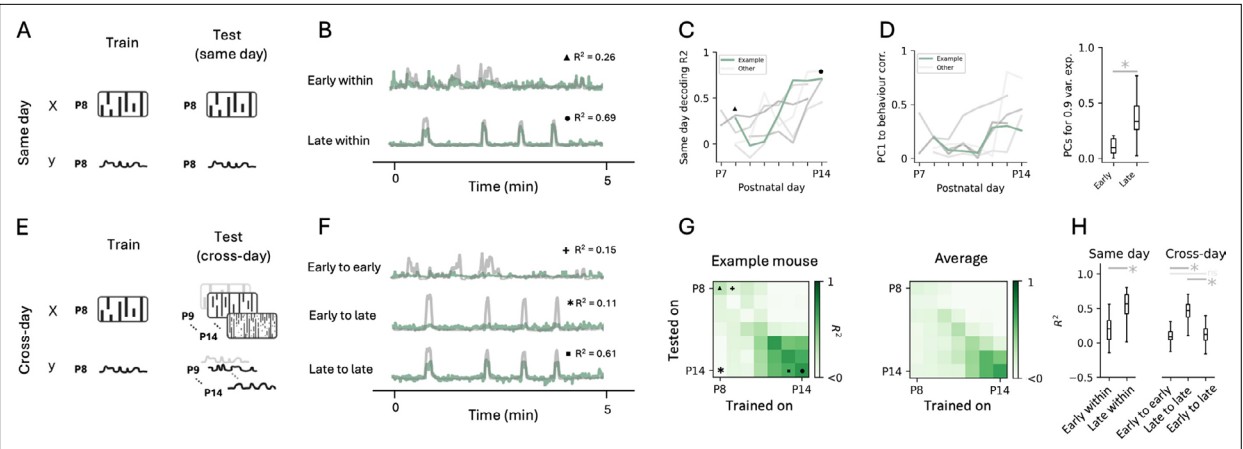

**Figure 7.** Regression analysis to study the development and stability of neural representations. (**A**) Left: A decoding model is fitted on each day to predict a behavioural variable (y, mouse motion) from the simultaneously recorded calcium imaging data (X, activity raster). Right: The model is then tested on the same day (using cross-validation) to assess the encoding of the variable on that given day. (**B**) Overlay of animal's motion (grey) and the predicted signal from neural activity (green) fit on the same day for example early (postnatal day 8 [P8], top) and late recordings (P14, bottom). Symbols indicate the corresponding $R^2$ values in panels C and G. Only the first 5 min of the recording are shown for visualisation purposes; for full traces of all recordings, see *Figure 7—figure supplement 1*. (**C**) Graphs indicating $R^2$ values for same-day cross-validated decoding performance as a function of mouse age. Green indicates the example mouse (also in D). (**D**) Graph plotting the correlation between the animal's motion and the first principal component signal across days (left) and a statistical comparison of the early (≤P11) and late (>P11) epochs (right). *: Mann-Whitney U test, p=1.2 × 10⁻³. (**E**) Left: Same as in A, an individual model is fitted on each day. Right: To assess the stability of the representation, each model is tested across different days, which is only possible when tracking neurons across sessions. (**F**) Same as B but for examples of cross-day decoding early to early (P9 to P8), late to late (P13 to P14), and early to late (P8 to P14). Symbols indicate the corresponding $R^2$ values in panel G. (**G**) Full prediction performance matrix showing $R^2$ values for all combinations of fit and test datasets (rows and columns, respectively) for the example mouse (left) and average across all mice for the same period (right). Diagonal entries correspond to same-day decoding, off-diagonal entries to cross-day recording. (**H**) Box plots comparing $R^2$ values for same-day (left) or across-day (right) decoding for early (≤P11) and late (>P11) developmental epochs pooled across all mice. *: Statistically significant; ns: not statistically significant (Kruskal-Wallis test: p=4.5 × 10⁻¹⁰; post hoc Mann-Whitney U tests with Bonferroni correction, 'early within' to 'late within': p=1.6 × 10⁻³, 'early to early' to 'late to late': p=6.5 × 10⁻⁶, 'early to early' to 'early to late': p=ns, 'late to late' to 'early to late': p=2.4 × 10⁻⁵; for all possible comparisons, see *Figure 7—figure supplement 1I*).

The online version of this article includes the following figure supplement(s) for figure 7:

**Figure supplement 1.** Relationship between spontaneous behaviour and neural activity for all mice in the dataset.

Hence, by combining Track2p longitudinal cell tracking with decoding approaches, one can systematically map the emergence and stability of cortical representations in the developing brain.

## Discussion

Here, we described Track2P, a cell tracking algorithm that can be used to follow the changing dynamics of hundreds of matched neurons from daily calcium imaging recordings in the growing brain of living mouse pups. At the core of the method lies sequential affine registration followed by cell matching for each subsequent pair of recordings, which leads to a better tracking performance during development when using Track2p compared to other methods. Using this benchmarked approach, we observed a sharp developmental transition from highly synchronised activity to multidimensional, behaviour state-dependent neural dynamics. Beyond this key finding, our method opens a much-needed tool for investigating the developmental functional trajectories of individual neurons during early postnatal brain development, and their deviations caused by genetic mutations or environmental perturbations.

Developing and benchmarking Track2p showed that it is able to track large populations of neurons despite substantial brain growth and other developmental changes through extended periods of time during development. We suggest this is mostly due to affine transform being a good approximation of the growth processes that occur between 2 consecutive days of imaging, with the cell matching step allowing to account for slight non-linearities in tissue growth day-by-day. Currently available methods typically register all imaging FOVs to a single reference, which we believe results in the accumulation of non-linearities, making registration more difficult and resulting in poorer tracking performance.

From a practical perspective, Track2p reduces manual cell tracking time by transforming a tedious process of manual annotation into a fast and automated procedure. Indeed, considering the time needed for manually tracking all cells in our dataset, we estimate that it would require approximately 5 person-days (120 hr) of work per subject under our experimental conditions, highlighting the necessity of automatic tracking in large-scale calcium imaging recordings. To aid future research, we provide Track2p fully open-source, with substantial documentation and an accompanying GUI facilitating the use of the algorithm by users without previous coding knowledge.

Besides developing the algorithm, we also showcase analysis techniques that can be used to gain unique insights from longitudinal calcium recordings, drawing inspiration from previous research studying plasticity, learning, and representational drift in the adult brain (*Ziv et al., 2013*; *Driscoll et al., 2017*; *Colonnese, 2014*). We specifically highlight: (1) quantifications of functional statistics of the tracked population across days (*Figure 4*); (2) correlation analyses to compare these across days (*Figure 5*); (3) regression analyses to study the emergence and stability of neural representations across recording days (*Figure 6*).

Last, we would like to emphasise that this analytical advancement is grounded on optimised experimental procedures to image daily cortical dynamics through a glass window mounted on the developing pup head. Previous studies had described adaptations of the surgical procedure or head fixation to developing pups (*Che and De Marco García, 2021*); however, by measuring the increase in pairwise distance between tracked neurons as a function of age per animal, we now provide a quantitative metric to estimate the invasiveness of the procedure. Interestingly, we observed a growth rate that closely matched ex vivo quantifications (*Thompson et al., 2014*), indicating a minimal impact of our experimental preparation on anatomical brain growth, also confirmed by mouse weight measurements. While we observed a slightly slower weight gain in imaged pups compared to their littermates, further investigation is needed to definitively assess whether repetitive daily imaging might influence cognitive developmental trajectories, e.g., through maternal separation.

Applying these techniques in our dataset during early postnatal development, we observed a significant and fast change in the nature of neuronal dynamics and state modulation in the barrel cortex, centred on P11, that manifests in several ways: (1) an increase in activity rates; (2) a decrease in pairwise correlations; (3) a shift from locally clustered to widely distributed correlations; (4) an increase in the dimensionality of spontaneous activity; (5) a remapping of the functional pairwise correlation structure; and (6) the emergence of a stable representation of spontaneous active movement.

If the first three changes had already been described previously in different cortical areas (*Wu et al., 2024*; *Golshani et al., 2009*; *Mizuseki and Buzsáki, 2013*; *Stringer et al., 2019a*; *Colonnese et al., 2010*; *Dard et al., 2022*; *Mòdol et al., 2024*; *Rio-Bermudez and Blumberg, 2021*), the last two could not be observed without the dense longitudinal tracking of the same neurons across several days in the growing brain permitted by Track2P. This multifaceted transition likely reflects several concurrent developmental processes, including changes in sleep architecture and neuromodulatory tone (*Dard et al., 2022*; *Ahmad et al., 2024*; *Marques-Smith et al., 2016*; *Anastasiades et al., 2016*), maturation and rewiring of local inhibitory circuits (*Bollmann et al., 2023*; *Modol et al., 2020*; *Colonnese et al., 2010*; *Rio-Bermudez and Blumberg, 2021*; *Tuncdemir et al., 2016*; *Chini et al., 2022*; *Gribizis et al., 2019*; *Minlebaev et al., 2011*) or disengagement from peripheral sensory inputs (*Dard et al., 2022*; *Shen and Colonnese, 2016*). This postnatal spurt is correlated with sparsification, disappearance of the spindle-burst oscillations, and increase in dimensionality of the representations (*Golshani et al., 2009*; *Mizuseki and Buzsáki, 2013*; *Stringer et al., 2019a*; *Colonnese et al., 2010*; *Dard et al., 2022*; *Micheva and Beaulieu, 1996*; *Luhmann and Khazipov, 2018*; *Blumberg et al., 2022*; *Tiriac et al., 2014*). Such drastic and fast changes are accompanied by significant behavioural changes, maybe the most salient ones being the change in the nature and duration of sleep, as well as the start of active exploration (*Tiriac and Blumberg, 2016*). The start of behavioural state modulation and dimensionality increase could reflect the same phenomenon by which spontaneous motor behaviour, including facial movements, drives multidimensional brainwide activity in the adult visual cortex (*Benisty et al., 2024*). This representation is present and stable from P11 onwards, as revealed using our decoding approach. The early absence of representation of spontaneous motor behaviour is in agreement with the previously reported lack of reafferent brain activity in response to self-generated wake movement until P11 (*Dooley and Blumberg, 2018*; *Khazipov et al., 2004*; *Inacio et al., 2023*). Thus, although we did not explicitly distinguish here

between sleep and wake-generated movements, it is likely that the 'motion energy' metric combined with the decoding mostly took into account wake behaviour, given the long periods of motor activity observed. Hence, we cannot exclude that twitching activity occurring during active sleep could contribute to an earlier representation of spontaneous motion, as reported previously (*Domínguez et al., 2021*). More refined behavioural analysis, particularly of sleep-wake transitions and twitches, could provide additional insights.

Therefore, the mid-second postnatal week marks a transition between two stable FC structures, indicating that globally, early and late functional connectivities differ. However, singular developmental trajectories cannot be excluded. Future studies could examine whether different cell types transition at different times or different paces, e.g., depending on their time of birth (*Cossart and Garel, 2022*), or whether unique neurons, such as hub cells, could maintain exceptional and stable FC (*Bollmann et al., 2023*). One population of particular interest for tracking is the neurons that contribute the most to behavioural prediction. Indeed, these could form a distinct population of movement-correlated neurons embedded in specific wiring schemes, as recently demonstrated (*Tasnim et al., 2024*).

This developmental transition centred around P11, which corresponds roughly to birth in human brain development, spans a period between 1 and 4 days, depending on the initial weight of the imaged pup. This transition may not yet be the last step before a mature adult-like network but instead represent a transient state (*Ratsifandrihamanana et al., 2023*) preceding the emergence of mature activity patterns. Its timing, correlated with animal weight, suggests it may represent a conserved developmental milestone. Hence, comparative studies across brain regions and species, including potential parallels in human development, would be valuable.

We believe that combining Track2p tracking and the analysis methods described here provides a template for future investigations of more complex developmental phenomena, such as the emergence of sensory representations and cognitive functions. Also, longitudinal imaging uniquely enables investigation of activity-dependent development, including how early activity patterns predict later functional properties and assembly formation of the same cells. Indeed, there is nothing precluding further studies to continue tracking the activity of the same cells until adulthood. Additionally, this approach opens possibilities for targeted manipulation studies to examine how early perturbations affect subsequent circuit development. The ability to track the same cells throughout early postnatal development should thus open doors to entirely new classes of experiments not possible before, providing deeper mechanistic insights into developmental principles and pathologies. This is even more important considering that alterations of developmental trajectories at early postnatal time points are starting to be pointed out as the roots for many developmental disorders (*Wu et al., 2024*; *Stringer et al., 2019b*).

## Resource availability

All code is available at the Track2p repository: https://github.com/juremaj/track2p (copy archived at *Mantez and Majnik, 2025*) with more extensive documentation and demos available at: https://track2p.github.io/home.html. The data, including preprocessed neural data for the tracked cells, behavioural data, and ground truth data for cell tracking, are available at: https://zenodo.org/records/17091226.

## Methods

### Data acquisition

#### Animals

All experimental procedures were approved by the French ethics committee (Ministère de l'Enseignement Supérieur, de la Recherche et de l'Innovation [MESRI]; Comité d'éthique CEEA-014; APAFiS # 30716-2021032215171216 v8) and conducted in agreement with the European Council Directive 86/609/EEC. GAD67-Cre mice were kindly donated by Dr. Hannah Monyer (Heidelberg University). Mice were bred and stored in an animal facility with room temperature (RT) and relative humidity maintained at 22 ± 1°C and 50 ± 20%, respectively. Mice were provided ad libitum access to water and food. A total of six mice were used in the study, all heterozygous GAD67-Cre transgenics.

## Virus injections

We performed viral injections at P0 as previously described (*Ntatsis et al., 2023*). Briefly, we prepared a solution of AAV-hSyn-GCaMP8m and AAV-FLEX-tdTomato (2:1 volumetric ratio, $10^{12}$ genome copies per millilitre; Addgene) with a small volume (10:1) of 0.05% trypan blue (T8154 Sigma) to verify the success of the injection. We then briefly anaesthetised the mouse on ice and injected 2 µL of the solution in the right lateral ventricle using a glass micropipette. Pups were left to recover on a heating pad at 37°C before being returned to the dam.

## Cranial window surgery

All procedures were performed as in *Bollmann et al., 2023*. Surgeries were performed at P7 for all mice. Briefly, betadine and lidocaine cream were applied topically, covering the area of the intended incision. Isoflurane was used for induction of global anaesthesia and maintained via a nose cone throughout the procedure. A heating pad was used to maintain body temperature. After skin removal, a head plate (4 mm inner diameter, Luigs and Neumann) was fixed to the part of the skull covering the barrel cortex using glue (SuperGlue3) and Super Bond (DSM Dentaire). A craniotomy was performed within the head plate opening before finally applying a thin layer of Kwik-Sil (WPI) to the surface of the dura and covering it with a glass cover slip (3 mm, Warner Instruments). The cover slip was last fixed to the headplate and the skull again using glue and Super Bond. Mice were left to recover on a heating pad at 37°C for at least 1 hr before returning them to the home cage.

## Chronic two-photon calcium imaging

Longitudinal two-photon calcium imaging was performed for each mouse for at least 6 consecutive days (see *Figure 5* for details). Imaging was performed using a Bruker (Ultima 2P) microscope with a Coherent Mai-Tai excitation laser (950 nm excitation light) and a 16× Nikon objective (NA 0.8). Before detection, emitted light was split into two optical paths, each associated with a filter (red and green, 580–620 and 500–550 nm, respectively), allowing us to simultaneously record the GCaMP8m and the tdTomato signal. The acquisition was performed using the Prairie View software. All recordings were performed in layer 2/3 (depth between 100 and 200 µm from the pial surface) with a 720×720 µm² FOV and 512×512 pixel resolution. Imaging rate was 30 Hz (resonant scanner) and each session lasted 20 min. All experiments were performed in the dark, under sensory-minimised conditions, with mice being free to spontaneously run on a non-motorised treadmill (Luigs and Neumann). To facilitate alignment and cell tracking, we kept the alignment of the head mount with respect to the microscope fixed across all sessions for the same mouse. To record from the same region across days, we manually aligned the imaging plane in x, y, and z to best match the reference images of the red channel (tdTomato) from all previous recording days. During the course of each recording, pups were kept warm by a heating element mounted to the imaging setup. After each imaging session, the pups were returned to their home cage with the dam and their littermates.

## Videography

All videography was performed using a Basler camera (Basler ACE2 1920), with an infrared LED light source (ThorLabs 850 nm) pointed at the mouse. Videos were recorded at 30 Hz, with the microscope acquisition acting as a trigger for camera frame acquisition, also allowing for simple synchronisation across the two modalities. All acquisition was done using custom Python scripts using the PyPylon wrapper for the Basler camera software (Pylon Camera Software Suite, https://github.com/basler/pypylon; *Möller et al., 2025*).

## Processing of calcium imaging data

Calcium imaging data was preprocessed using the Suite2p pipeline, sequentially performing motion correction, ROI detection, signal extraction, and spike deconvolution (*Otsu, 1979*) for each recording separately. Suite2p additionally provides a cell classification feature, providing a probability of a classifier categorising an ROI as being a true cell. We considered all ROIs above the default threshold of 0.5 as true cells. We used baseline-corrected fluorescence traces as our dF/F (using the default Suite2p parameters) for all subsequent analyses. To facilitate the use of Track2p with other preprocessing pipelines, we implemented a data loader that takes as input three simple NumPy arrays

corresponding to ROIs, FOV, and traces (for more information, see https://track2p.github.io/run_inputs_and_parameters).

## Preprocessing videography

We used the global movements of the mouse as a proxy of its arousal state. Similarly to *Crouse, 2016*, we quantified these by looking at the pixel-wise difference of consecutive frames in the videography recording. Namely, we first took each two consecutive frames and computed their pixel-wise difference. We then squared all individual pixel-wise values and summed across pixels. This yielded a scalar value quantifying the motion of the mouse at each time point, which was used for all subsequent analyses.

## Cell tracking

### Image and ROI registration

As explained in the main text, Track2p aligns the ROIs based on mean FOV image registration. The implementation allows the user to choose which channel to use for computing the transformation ('anatomical' or 'functional' channel), as well as which types of transformation to estimate (rigid or affine). Unless otherwise stated, we used the 'anatomical' (tdTomato in GAD67-Cre expressing cells) channel with affine transformation (referred to as the 'baseline' condition). For the purposes of evaluation, we additionally ran tracking using the functional channel with affine registration ('functional' in *Figure 4*) and using the anatomical channel with rigid registration ('rigid' in *Figure 4*). Once registering the images, we applied the same transform to all ROIs from the subsequent session that were considered as cells (see main text). All Track2p image registration is done using the itk-elastix toolbox (*Varoquaux et al., 2017*).

### Cell matching

Cell matching was done in a way to maximise the sum of the IoU across all matches of ROIs between two sessions. For this reason, we defined a cost matrix M with entries corresponding to:

$$M_{i,j} = 1 - IoU\left(r_{sk,i}, r'_{s_{k+1},j}\right)$$

where $r_{s\_k,\,i}$ is the ith ROI in session $s_k$ and $r'_{s\_(k+1),\,j}$ is the jth ROI in the subsequent session $s_{k+1}$ after registering it to the coordinate system of $s_k$. Assigning matches between the two sets of ROIs in this way corresponds to a linear sum assignment problem, with the goal to find a matrix X that yields the optimal assignment cost:

$$\min_X \sum_{i=1}^{m_s} \sum_{j=1}^{m_s+1} M_{i,j} X_{i,j}$$

where $X_{i,j}$ equals 1 if and only if ROI $r_{s,i}$ is assigned to $r_{s+1,i}$ and 0 otherwise. Additionally, since the number of ROIs is not necessarily the same across two sessions, M and X are not necessarily square. In this case, if there are more rows than columns, then not every row needs to be assigned to a column, and vice versa. We solve this problem by using the algorithm described in *Glaser et al., 2020*, and implemented in SciPy.

Additionally, some of the matches can be putative false positives. To avoid this issue, we use an approach similar to the one described in *Tolu et al., 2010*, using a statistical threshold to filter matches based on their IoU distribution. To do this, we use the Skimage implementation of Otsu's method applied to the IoU histogram (*Ulman et al., 2017*).

### Generating a ground truth dataset

Since manually tracking all cells would require prohibitive amounts of time (see Discussion), we decided to generate sparse manual annotations, only tracking a subset of all cells from the first recording day onwards. To do this, we took the first recording ($s_0$), and we defined a grid of 8×8 (64) equidistant points over the FOV and, for each point, identified the closest ROI in terms of Euclidean distance from the median pixel of the ROI (see *Figure 4—figure supplement 1A*). We then manually tracked these 64 ROIs across subsequent days. The manual tracking was done using the Suite2p GUI, by

visualising the FOV and masks from two recordings side by side and choosing the matching ROI from the subsequent recording or terminating the track if we could not find a match (see *Figure 4—figure supplement 1B*). Only neurons that were detected and tracked across all sessions were taken into account and referred to as our ground truth dataset ('GT' in *Figure 4*). When comparing the GT to Track2p tracks, we only considered Track2p tracks that originated from one of the 64 ROIs chosen for evaluation initially. All manual tracking was performed blind to the Track2p outputs.

## Tracking evaluation metrics

For the evaluation of Track2p under different conditions and comparison to CellReg, we used the 'Complete tracks' metric (*Sheintuch et al., 2017*; *Rochefort et al., 2009*, *Figure 3B*), defined as:

$$CT = \frac{2 \cdot T_{rc}}{T_c + T}$$

where $T_{rc}$ is the number of perfectly reconstructed tracks, $T_c$ is the number of total tracks identified by Track2p for the 64 $s_0$ ROIs chosen for manual tracking, and $T_{gt}$ is the number of total tracks in the ground truth dataset (from the same 64 $s_0$ ROIs). In the case of perfect tracking, CT will be equal to 1 (where all computed tracks are equivalent to the ground truth tracks [$T_{rc} = T_c = T_{gt}$]). Conversely, in the case of failed tracking, the value of CT will be close to 0 (e.g. when there is a poor match between ground truth and reconstructed tracks [$T_{rc} \approx 0$] or when there are many falsely reconstructed tracks not present in the ground truth [$T_c \gg T_{gt}$]). The CT metric is mathematically equivalent to an F1 score where true positives are defined as perfectly reconstructed tracks (*Figure 4A*, row 1), false negatives as tracks from GT without a match in Track2p (*Figure 4A*, row 2) and false positives as tracks from Track2p without a match in GT (*Figure 4A*, rows 3, 4, and 5).

In the final step of evaluation, we looked at when the algorithms from the original evaluation failed in their tracking of the ground truth neurons. For this, we used a metric of the proportion of correctly reconstructed tracks for an increasingly longer number of sessions (s) (referred to as 'Prop. correct' in *Figure 4C–F*):

$$T_{rc}\left(s\right) = \frac{T_{rc}\left(s\right)}{T}$$

where $T_{gt}$ is equivalent as above and $T_{rc}(s)$ is equivalent as $T_{rc}$ but for full tracks up to session s. Since in this case we kept the GT consistent with the initial evaluation, including only the neurons identified across all days, this metric is agnostic to possible false positives at shorter time epochs.

To facilitate evaluation by individual users under different experimental conditions, we provide a helper script (Jupyter notebook) aiding the whole evaluation process by both defining a grid of cells to manually track (see previous section), as well as to compute the tracking quality metrics once the ground truth dataset is completed.

## CellReg tracking

To compare Track2p with CellReg, we ran the MATLAB implementation of CellReg tracking (https://github.com/zivlab/CellReg; *Sheintuch, 2024*) according to the provided user manual. We first attempted tracking using the default rigid registration, which yielded an error, terminating the algorithm and indicating that subsequent sessions do not resemble the reference session and suggesting to use non-rigid registration. We then re-launched the algorithm with non-rigid registration and used those results to evaluate the tracking the same way as for Track2p (see Results). Notably, even when running CellReg using non-rigid registration, we noticed that the algorithm did not find any tracks spanning all days ($T_{rc} = 0$), which explains the CT score of 0 for all day evaluation.

## Functional properties of tracked neurons

### Calcium event rates

To quantify the calcium event rate statistics, we used SciPy's peak detection algorithm (Python). For this, we first denoised the traces slightly by averaging using a bin size of 10 frames and then proceeded to detect any peaks with a height and prominence of at least one standard deviation. We then calculated the rate as the number of peaks per minute within the recording. When quantifying the stability

of the rates across days, we computed the Pearson correlation coefficient across all neurons for each possible combination of sessions recorded from the same mouse.

## Pairwise correlations and PCA

We quantified pairwise correlations by computing the Pearson correlation between the full traces of all pairs of simultaneously recorded neurons within each session. To quantify the spatial properties of pairwise correlations, we additionally calculated the Euclidean distance between ROI centroids for each corresponding pair of neurons. When plotting the pairwise correlations as a function of pairwise distance, we averaged in bins of 30 µm. The correlation of neighbouring neurons was estimated as the intercept of an exponential fit to the full data.

When comparing the stability of the correlation structure across days, we refer to this as 'FC stability', which we calculate as the Pearson correlation between all pairs of original correlation matrices for a given mouse. When fitting a sigmoid using the first day as the reference, we fix the upper limit of the sigmoid to within-day FC stability for that day and leave the other parameters free. We calculate the inflection point ('transition age') and the linear portion ('transition time') of the sigmoid using the extrema of the first and fourth derivatives, respectively.

For PCA, we used the scikit-learn implementation (*Pedregosa et al., 2012*, Python).

## Decoding

All decoding was done using linear regression with ridge regularisation (ridge regression) to avoid overfitting given the large number of neurons. Ridge regression optimises the weights (β) that minimise the following loss function:

$$\mathcal{L}(\boldsymbol{\beta}) = \|\mathbf{y} - \mathbf{X}\boldsymbol{\beta}\|_2^2 + \lambda\|\boldsymbol{\beta}\|_2^2$$

where in our case y is behavioural data, X is neural data, and $\lambda$ is a regularisation parameter. To choose the optimal $\lambda$ and to reliably estimate the model performance on same-day decoding, we used nested cross-validation (see for example *Varoquaux et al., 2017*; *Glaser et al., 2020*), ensuring data efficiency and a splitting between training, validation, and test data. Briefly, a grid search for the optimal $\lambda$ is performed by using the training set to fit models with different $\lambda$ values and choosing the value corresponding to the model with the best performance on the held-out validation set (model selection). This model is then evaluated on the test data that was used neither in model fitting nor in the hyperparameter optimisation (model evaluation). To ensure data efficiency, nested cross-validation repeats this procedure for all possible splits of the data, with an outer cross-validation loop used for model evaluation and an inner loop for model selection. We used fivefold splits for both the inner and outer loops; splits were done on consecutive 2 min blocks of the recording. For cross-day decoding, we fit a new model with the optimal lambda for that given day and evaluated it on all other days. For all decoding analysis, we slightly denoised the dF/F, as well as the behaviour traces by averaging in bins of 10 consecutive timestamps. All models were implemented and fitted using PyTorch (Python).

## Acknowledgements

We thank all the members of the Cossart lab for helpful discussions and constructive feedback. We thank Drs. Lorenzo Fontolan and Michel Picardo for critical feedback on the manuscript. We thank INMED's animal facility and PBMC technological platform for excellent technical support. This work was supported by the European Research Council under the European Union's Horizon 2020 research and innovation programme (grant agreement no. 951330), by the Fondation Bettencourt Schueller and the Fondation Roger de Spoelberch; RC is supported by CNRS. J-CP is supported by INSERM. The project leading to this publication has received funding from the French government under the 'France 2030' investment plan managed by the French National Research Agency (reference: ANR-16-CONV000X/ANR-17-EURE-0029) and from Excellence Initiative of Aix-Marseille University – A*MIDEX (AMX-19-IET-004).

## Additional information

### Funding

| Funder | Grant reference number | Author |
|---|---|---|
| European Research Council | 951330 | Rosa Cossart |
| Fondation Bettencourt Schueller | | Rosa Cossart |
| Fondation Roger de Spoelberch | | Rosa Cossart |
| Agence Nationale de la Recherche | ANR-16-CONV000X / ANR-17-EURE-0029 | Rosa Cossart |
| Aix-Marseille Université | A*MIDEX (AMX-19-IET-004) | Rosa Cossart |

The funders had no role in study design, data collection and interpretation, or the decision to submit the work for publication.

### Author contributions

Jure Majnik, Conceptualization, Resources, Data curation, Software, Formal analysis, Investigation, Methodology, Writing – original draft, Writing – review and editing; Manon Mantez, Software, Writing – original draft; Sofia Zangila, Investigation, Methodology; Stéphane Bugeon, Methodology; Léo Guignard, Methodology, Writing – original draft; Jean-Claude Platel, Conceptualization, Supervision, Writing – original draft, Writing – review and editing; Rosa Cossart, Conceptualization, Supervision, Funding acquisition, Writing – original draft, Writing – review and editing

### Author ORCIDs

Jure Majnik (ID) https://orcid.org/0009-0005-6296-3659
Manon Mantez (ID) https://orcid.org/0009-0000-6376-9704
Sofia Zangila (ID) https://orcid.org/0009-0002-4378-2797
Léo Guignard (ID) https://orcid.org/0000-0002-3686-1385
Jean-Claude Platel (ID) https://orcid.org/0000-0001-5542-3076
Rosa Cossart (ID) https://orcid.org/0000-0003-2111-6638

### Ethics

All experimental procedures were approved by the French ethics committee (Ministère de l'Enseignement supérieur, de la Recherche et de l'Innovation (MESRI); Comité d'éthique CEEA-014; APAFiS # 30716-2021032215171216 v8) and conducted in agreement with the European Council Directive 86/609/EEC. GAD67-Cre mice were kindly donated by Dr. Hannah Monyer (Heidelberg University). Mice were bred and stored in an animal facility with room temperature (RT) and relative humidity maintained at 22 ± 1°C and 50 ± 20%, respectively. Mice were provided ad libitum access to water and food. A total of 6 mice were used in the study, all heterozygous GAD67-Cre transgenics.

Reviewer #1 (Public review): https://doi.org/10.7554/eLife.107540.3.sa1
Reviewer #2 (Public review): https://doi.org/10.7554/eLife.107540.3.sa2
Reviewer #3 (Public review): https://doi.org/10.7554/eLife.107540.3.sa3
Author response https://doi.org/10.7554/eLife.107540.3.sa4

## Additional files

### Supplementary files
MDAR checklist

### Data availability
All code is available at the Track2p repository: https://github.com/juremaj/track2p (copy archived at *Mantez and Majnik, 2025*) with more extensive documentation and demos available at: https://

track2p.github.io/home.html. The data including pre-processed neural data for the tracked cells, behavioural data and ground truth data for cell tracking are available at: https://doi.org/10.5281/zenodo.17091226.

The following dataset was generated:

| Author(s) | Year | Dataset title | Dataset URL | Database and Identifier |
|---|---|---|---|---|
| Majnik J, Mantez M, Zangila S, Bugeon S, Guignard L, Platel JC, Cossart R | 2025 | Longitudinal tracking of neuronal activity from the same cells in the developing brain using Track2p | https://doi.org/10.5281/zenodo.17091226 | Zenodo, 10.5281/zenodo.17091226 |

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

## Appendix 1

### Interactive visualisation and curation using the Track2p GUI

To facilitate ease of use and to visually evaluate and curate the quality of Track2p tracking, we've designed an intuitive GUI that makes it easy for users to launch the Track2p algorithm and interact with the outputs. Track2p GUI allows the users to perform cell tracking, as well as generate some preliminary visualisations of cell activity without requiring any programming skills on behalf of the user (*Figure 2—figure supplement 1A*). Additionally, we've made the algorithm and the GUI easy to install via pip, and provide plenty of documentation on the project's website: https://track2p. github.io/home.html.

Track2p provides several data visualisation functionalities. Firstly, while running the algorithm, a set of figures is generated that can be used to monitor the progress at each consecutive step, as well as to provide a visual overview of tracking quality, e.g., based on a visual assessment of registration quality, IoU histograms, or zoomed-in images of example tracked cells. The GUI itself provides an interactive view of the FOV overlaid with ROIs tracked across all days (*Figure 2—figure supplement 1B*, similar to *Figure 3D*). The user can then select a cell from this window that is then displayed in a zoomed-in view across all days (*Figure 2—figure supplement 1B*, similar to *Figure 3C*). The visualised cell can be selected interactively by selecting it from contours displayed over the whole FOV. Based on these visualisations, users can also assess the tracking performance and, if necessary, manually curate the results by manually identifying cases of incorrect tracking (*Figure 2—figure supplement 1E*). Under conditions shown in this paper, manual curation should not be strictly necessary (see *Figure 4* for evaluation without manual curation); however, this will depend greatly on the particularities of the dataset used.

Additionally, the GUI also provides activity visualisation, plotting the activity time series of the selected ROI for each recording (*Figure 2—figure supplement 1C*, similar to *Figure 5—figure supplement 1B*), while also offering the visualisation of the activity of the whole population across all days by generating raster plots of neuronal activity (*Figure 2—figure supplement 1F*, similar to *Figure 5A*, *Figure 5—figure supplement 1A*). These visualisations can allow the user to visually inspect changes to activity statistics at the single-cell level, as well as changes to population-level phenomena such as synchronisation (assemblies) and sequences at the level of all tracked neurons. Raster sorting capabilities can also give a visual insight into the stability of these phenomena across recordings. Currently, the user can choose between PCA or t-SNE (t-distributed stochastic neighbour embedding) techniques to re-sort the rows of the raster based on dimensionality reduction. These methods aim to sort cells with similar fluorescence traces in adjacent rows of the raster plot, visualising co-varying activity patterns at the population level. Additionally, the user can choose between individual sorting and sorting that is preserved across days. Individual sorting better shows these population phenomena for each specific day; however, in this case, the correspondence between the cells across days is lost due to independent sorting. On the other hand, across-day sorting allows the user to focus on 1 day to visualise the stability of these population phenomena and see if they are preserved across other days, since the sorting of the neurons remains consistent across sessions.

