## [Editor Report · eLife Assessment]

This **fundamental** study presents a new method for longitudinally tracking cells in two-photon imaging data that addresses the specific challenges of imaging neurons in the developing cortex. It provides **compelling** evidence demonstrating reliable longitudinal identification of neurons across the second postnatal week in mice. The study should be of interest to development neuroscientists engaged in population-level recordings using two-photon imaging.

---

## [Referee Report · Reviewer #1 (Public review)]

Summary:

This manuscript presents a compelling and innovative approach that combines Track2p neuronal tracking with advanced analytical methods to investigate early postnatal brain development. The work provides a powerful framework for exploring complex developmental processes such as the emergence of sensory representations, cognitive functions, and activity-dependent circuit formation. By enabling the tracking of the same neurons over extended developmental periods, this methodology sets the stage for mechanistic insights that were previously inaccessible.

Strengths:

(1) Innovative Methodology:

The integration of Track2p with longitudinal calcium imaging offers a unique capability to follow individual neurons across critical developmental windows.

(2) High Conceptual Impact:

The manuscript outlines a clear path for using this approach to study foundational developmental questions, such as how early neuronal activity shapes later functional properties and network assembly.

(3) Future Experimental Potential:

The authors convincingly argue for the feasibility of extending this tracking into adulthood and combining it with targeted manipulations, which could significantly advance our understanding of causality in developmental processes.

(4) Broad Applicability:

The proposed framework can be adapted to a wide range of experimental designs and questions, making it a valuable resource for the field.

Weaknesses:

None major. The manuscript is conceptually strong and methodologically sound. Future studies will need to address potential technical limitations of long-term tracking, but this does not detract from the current work's significance and clarity of vision

Comments on revisions:

I have no further requests. I think this is an excellent manuscript

---

## [Referee Report · Reviewer #2 (Public review)]

Summary:

The manuscript by Majnik and colleagues introduces "Track2p", a new tool designed to track neurons across imaging sessions of two-photon calcium imaging in developing mice. The method addresses the challenge of tracking cells in the growing brain of developing mice. The authors showed that "Track2p" successfully tracks hundreds of neurons in the barrel cortex across multiple days during the second postnatal week. This enabled identification of the emergence of behavioral state modulation and desynchronization of spontaneous network activity around postnatal day 11.

Strengths

The authors have satisfactorily addressed the majority of our questions and comments, and the revisions substantially improve the manuscript. The expansion of Track2p to accept general NumPy array inputs makes the tool more accessible to researchers using different analysis pipelines. While the absence of benchmarking standards remains a limitation across the field, the release of the ground-truth dataset is an important step forward that will allow other researchers to evaluate and compare algorithms.

Minor point

(1) The authors tested the robustness of the algorithm across non-consecutive days. As expected, performance drops significantly under these conditions. We agree that this limitation reflects biological constraints due to brain growth rather than shortcomings of the algorithm itself. This is relevant for researchers planning to use Track2p for longitudinal imaging or benchmarking new algorithms, and we recommend including some of this information in the Supplementary Information along with a brief discussion.

Comments on revisions:

We acknowledge the extended documentation for using Track2p and converting between Suite2p outputs and NumPy arrays. This addition is of great utility. We would also suggest further expanding the documentation for the NumPy array implementation, as we ran into some errors when testing this feature using NumPy arrays generated from deltaF traces, TIFF FOVs, and Cellpose masks.

---

## [Referee Report · Reviewer #3 (Public review)]

Summary:

In this manuscript Majnik et al. developed a computational algorithm to track individual developing interneurons in the rodent cortex at postnatal stages. Considerable development in cortical networks takes place during the first postnatal weeks, however, tools to study them longitudinally at a single cell level are scarce. This paper provides a valuable approach to study both single cell dynamics across days and state-drive network changes. The authors used Gad67Cre mice together with virally introduced TdTom to track interneurons based on their anatomical location in the FOV and AAVSynGCaMP8m to follow their activity across the second postnatal week, a period during which the cortex is known to undergo marked decorrelation in spontaneous activity. Using Track2P, the authors show feasibility to track populations of neurons in the same mice capturing with their analysis previously described developmental decorrelation and uncovering stable representations of neuronal activity, coincident with the onset of spontaneous active movement. The quality of the imaging data is compelling, and the computational analysis is thorough, providing a widely applicable tool for the analysis of emerging neuronal activity in the cortex. Below are some points for the authors to consider.

Major points

The authors use a viral approach to label cortical interneurons. It is unclear how Track2P will perform in dense networks of excitatory cells using GCaMP transgenic mice.

The authors used 20 neurons to generate a ground truth data set. The rational for this sample size is unclear. Figure 1 indicates capability to track ~728 neurons. A larger ground truth data set will increase the robustness of the conclusions.

It is unclear how movement was scored in the analysis shown in Fig 5A. Was the time that the mouse spent moving scored after visual inspection of the videos? Were whisker and muscle twitches scored as movement or was movement quantified as amount of time in which the treadmill was displaced?

The rational for binning the data analysis in early P11 is unclear. As the authors acknowledged, it is likely that the decoder captured active states from P11 onwards. Because active whisking begins around P14, it is unlikely to drive this change in network dynamics at P11. Does pupil dilation in the pups change during locomotor and resting states? Does the arousal state of the pups abruptly change at P11?

Comments on revisions:

The authors have addressed carefully all my comments. This is an interesting paper.

---

## [Author Response]

The following is the authors’ response to the original reviews.

**Reviewer #1 (Public review):**

We thank the reviewer for very enthusiastic and supportive comments on our manuscript.

Summary:This manuscript presents a compelling and innovative approach that combines Track2p neuronal tracking with advanced analytical methods to investigate early postnatal brain development. The work provides a powerful framework for exploring complex developmental processes such as the emergence of sensory representations, cognitive functions, and activity-dependent circuit formation. By enabling the tracking of the same neurons over extended developmental periods, this methodology sets the stage for mechanistic insights that were previously inaccessible.Strengths:(1) Innovative Methodology:The integration of Track2p with longitudinal calcium imaging offers a unique capability to follow individual neurons across critical developmental windows.(2) High Conceptual Impact:The manuscript outlines a clear path for using this approach to study foundational developmental questions, such as how early neuronal activity shapes later functional properties and network assembly.(3) Future Experimental Potential:The authors convincingly argue for the feasibility of extending this tracking into adulthood and combining it with targeted manipulations, which could significantly advance our understanding of causality in developmental processes.(4) Broad Applicability:The proposed framework can be adapted to a wide range of experimental designs and questions, making it a valuable resource for the field.Weaknesses:No major weaknesses were identified by this reviewer. The manuscript is conceptually strong and methodologically sound. Future studies will need to address potential technical limitations of long-term tracking, but this does not detract from the current work's significance and clarity of vision.
**Reviewer #2 (Public review):**
Summary:The manuscript by Majnik and colleagues introduces "Track2p", a new tool designed to track neurons across imaging sessions of two-photon calcium imaging in developing mice. The method addresses the challenge of tracking cells in the growing brain of developing mice. The authors showed that "Track2p" successfully tracks hundreds of neurons in the barrel cortex across multiple days during the second postnatal week. This enabled the identification of the emergence of behavioral state modulation and desynchronization of spontaneous network activity around postnatal day 11.Strengths:The manuscript is well written, and the analysis pipeline is clearly described. Moreover, the dataset used for validation is of high quality, considering the technical challenges associated with longitudinal two-photon recordings in mouse pups. The authors provide a convincing comparison of both manual annotation and "CellReg" to demonstrate the tracking performance of "Track2p". Applying this tracking algorithm, Majnik and colleagues characterized hallmark developmental changes in spontaneous network activity, highlighting the impact of longitudinal imaging approaches in developmental neuroscience. Additionally, the code is available on GitHub, along with helpful documentation, which will facilitate accessibility and usability by other researchers.Weaknesses:(1) The main critique of the "Track2p" package is that, in its current implementation, it is dependent on the outputs of "Suite2p". This limits adoption by researchers who use alternative pipelines or custom code. One potential solution would be to generalize the accepted inputs beyond the fixed format of "Suite2p", for instance, by accepting NumPy arrays (e.g., ROIs, deltaF/F traces, images, etc.) from files generated by other software. Otherwise, the tool may remain more of a useful add-on to "Suite2p" (see https://github.com/MouseLand/suite2p/issues/933) rather than a fully standalone tool.

We thank the reviewer for this excellent suggestion.

We have now implemented this feature, where Track2p is now compatible with ‘raw’ NumPy arrays for the three types of inputs. For more information, please check the updated documentation: https://track2p.github.io/run_inputs_and_parameters.html#raw-npy-arrays. We have also tested this feature using a custom segmentation and trace extraction pipeline using Cellpose for segmentation.

(2) Further benchmarking would strengthen the validation of "Track2p", particularly against "CaIMaN" (Giovannucci et al., eLife, 2019), which is widely used in the field and implements a distinct registration approach.

This reviewer suggested further benchmarking of Track2P. Ideally, we would want to benchmark Track2p against the current state-of-the-art method. However, the field currently lacks consensus on which algorithm performs best, with multiple methods available including CaIMaN, SCOUT (Johnston et al. 2022), ROICaT (Nguyen et al. 2023), ROIMatchPub (recommended by Suite2p documentation and recently used by Hasegawa et al. 2024), and custom pipelines such as those described by Sun et al. 2025. The absence of systematic benchmarking studies—particularly for custom tracking pipelines—makes it impossible to identify the current state-of-the-art for comparison with Track2p. While comparing Track2p against all available methods would provide comprehensive evaluation, such an analysis falls beyond the scope of this paper.

We selected CellReg for our primary comparison because it has been validated under similar experimental conditions—specifically, 2-photon calcium imaging in developing hippocampus between P17-P25 (Wang et al. 2024)—making it the most relevant benchmark for our developmental neocortex dataset.

That said, to support further benchmarking in mouse neocortex (P8-P14), we will publicly release our ground truth tracking dataset.

(3) The authors might also consider evaluating performance using non-consecutive recordings (e.g., alternate days or only three time points across the week) to demonstrate utility in other experimental designs.

Thank you for your suggestion. We have performed a similar analysis prior to submission, but we decided against including it in the final manuscript, to keep the evaluation brief and to not confuse the reader with too many different evaluation methods. We have included the results inAuthor response images 1 and 2 below.

To evaluate performance in experimental designs with larger time spans between recordings (>1 day) we performed additional evaluation of tracking from P8 to each of the consecutive days while omitting the intermediate days (e. g. P8 to P9, P8 to P10 … P8 to P14). The performance for the three mice from the manuscript is shown below:

**Author response image 1. sa4fig1:** 

As expected with increasing time difference between the two recordings the performance drops significantly (dropping to effectively zero for 2 out of 3 mice). This could also explain why CellReg struggles to track cells across all days, since it takes P8 as a reference and attempts to register all consecutive days to that time point before matching, instead of performing registration and matching in consecutive pairs of recordings (P8-P9, P9-P10 … P13-P14) as we do.

Finally for one of the three mice we also performed an additional test where we asked how adding an additional recording day might rescue the P8-P14 tracking performance. This corresponds to the comment from the reviewer, answering the question if we can only perform three days of recording which additional day would give the best tracking performance.

**Author response image 2. sa4fig2:** 

As can be seen from the plot, adding the P10 or P11 recording shows the most significant improvement to the tracking performance, however the performance is still significantly lower than when including all days (see Fig. 4). This test suggests that including a day that is slightly skewed to earlier ages might improve the performance more than simply choosing the middle day between the two extremes. This would also be consistent with the qualitative observation that the FOV seems to show more drastic day-to-day changes at earlier ages in our recording conditions.

**Reviewer #3 (Public review):**
Summary:In this manuscript, Majnik et al. developed a computational algorithm to track individual developing interneurons in the rodent cortex at postnatal stages. Considerable development in cortical networks takes place during the first postnatal weeks; however, tools to study them longitudinally at a single-cell level are scarce. This paper provides a valuable approach to study both single-cell dynamics across days and state-driven network changes. The authors used Gad67Cre mice together with virally introduced TdTom to track interneurons based on their anatomical location in the FOV and AAVSynGCaMP8m to follow their activity across the second postnatal week, a period during which the cortex is known to undergo marked decorrelation in spontaneous activity. Using Track2P, the authors show the feasibility of tracking populations of neurons in the same mice, capturing with their analysis previously described developmental decorrelation and uncovering stable representations of neuronal activity, coincident with the onset of spontaneous active movement. The quality of the imaging data is compelling, and the computational analysis is thorough, providing a widely applicable tool for the analysis of emerging neuronal activity in the cortex. Below are some points for the authors to consider.

We thank the reviewer for a constructive and positive evaluation of our MS.

Major points:(1) The authors used 20 neurons to generate a ground truth dataset. The rationale for this sample size is unclear. Figure 1 indicates the capability to track ~728 neurons. A larger ground truth data set will increase the robustness of the conclusions.

We think this was a misunderstanding of our ground truth dataset analysis which included 192 and not 20 neurons. Indeed, as explained in the methods section, since manually tracking all cells would require prohibitive amounts of time, we decided to generate sparse manual annotations, only tracking a subset of all cells from the first recording day onwards. To do this, we took the first recording (s0), and we defined a grid 64 equidistant points over the FOV and, for each point, identified the closest ROI in terms of euclidean distance from the median pixel of the ROI (see Fig. S3A). We then manually tracked these 64 ROIs across subsequent days. Only neurons that were detected and tracked across all sessions were taken into account and referred to as our ground truth dataset (‘GT’ in Fig. 4). This was done for 3 mice, hence 3X64 neurons and not 20 were used to generate our GT dataset.

(2) It is unclear how movement was scored in the analysis shown in Figure 5A. Was the time that the mouse spent moving scored after visual inspection of the videos? Were whisker and muscle twitches scored as movement, or was movement quantified as the amount of time during which the treadmill was displaced?

Movement was scored using a ‘motion energy’ metric as in Stringer et al. 2019 (V1) or Inácio et al. 2025 (S1). This metric takes each two consecutive frames of the videography recordings and computes the difference between them by summing up the square of pixelwise differences between the two images. We made the appropriate changes in the manuscript to further clarify this in the main text and methods in order to avoid confusion.

Since this metric quantifies global movements, it is inherently biased to whole-body movements causing more significant changes in pixel values around the whole FOV of the camera. Slight twitches of a single limb, or the whisker pad would thus contribute much less to this metric, since these are usually slight displacements in a small region of the camera FOV. Additionally, comparing neural activity across all time points (using correlation or R^2^) also favours movements that last longer (such as wake movements / prolonged periods of high arousal) since each time point is treated equally.

As we suggested in the discussion, in further analysis it would be interesting to look at the link between twitches and neural activity, but this would likely require extensive manual scoring. We could then treat movements not as continuous across all time-points, but instead using event-based analysis for example peri-movement time histograms for different types of movements at different ages, which is however outside of the scope of this study.

(3) The rationale for binning the data analysis in early P11 is unclear. As the authors acknowledged, it is likely that the decoder captured active states from P11 onwards. Because active whisking begins around P14, it is unlikely to drive this change in network dynamics at P11. Does pupil dilation in the pups change during locomotor and resting states? Does the arousal state of the pups abruptly change at P11?

We agree that P11 does not match any change in mouse behavior that we have been able to capture. However, arousal state in mice does change around postnatal day 11. This period marks a transition from immature, fragmented states to more organized and regulated sleep-wake patterns, along with increasing influence from neuromodulatory and sensory systems. All of these changes have been recently reviewed in Wu et al. 2024 (see also Martini et al. 2021). In addition, in the developing somatosensory system, before postnatal day 11 (P11), wake-related movements (reafference) are actively gated and blocked by the external cuneate nucleus (ECN, Tiriac et al. 2016 and all excellent recent work from the Blumberg lab). This gating prevents sensory feedback from wake movements from reaching the cortex, ensuring that only sleep-related twitches drive neural responses. However, around P11, this gating mechanism abruptly lifts, enabling sensory signals from wake movements to influence cortical processing—signaling a dramatic developmental shift from Wu et al. 2024

**Reviewer #1 (Recommendations for the authors):**
This manuscript represents a significant advancement in the field of developmental neuroscience, offering a powerful and elegant framework for longitudinal cellular tracking using the Track2p method combined with robust analytical approaches. The authors convincingly demonstrate that this integrated methodology provides an invaluable template for investigating complex developmental processes, including the emergence of sensory representations and higher cognitive functions.A major strength of this work is its emphasis on the power of longitudinal imaging to illuminate activity-dependent development. By tracking the same neurons over time, the authors open up new possibilities to uncover how early activity patterns shape later functional outcomes and the organization of neuronal assemblies-insights that would be inaccessible using conventional cross-sectional designs.Importantly, the manuscript highlights the potential for this approach to be extended even further, enabling continuous tracking into adulthood and thus offering an unprecedented window into long-term developmental trajectories. The authors also underscore the exciting opportunity to incorporate targeted perturbation experiments, allowing researchers to causally link early circuit dynamics to later outcomes.Given the increasing recognition that early postnatal alterations can underlie the etiology of various neurodevelopmental disorders, this work is especially timely. The methods and perspectives presented here are poised to catalyze a new generation of developmental studies that can reveal mechanistic underpinnings of both typical and atypical brain development.In summary, this is a technically impressive and conceptually forward-looking study that sets the stage for transformative advances in developmental neuroscience.

Thank you for the thoughtful feedback—it's greatly appreciated!

**Reviewer #2 (Recommendations for the authors):**
Minor points:(1) Figure 1. Consider merging or moving to Supplemental, as its rationale is well described in the text.

We would like to retain the current figure as we believe it provides an effective visual illustration of our rationale that will capture readers' attention and could serve as a valuable reference for others seeking to justify longitudinal tracking of the developing brain. We hope the reviewer will understand our decision.

(2) Some axis labels and panels are difficult to read due to small font sizes (e.g. smaller panels in Figures 5-7).

Modified, thanks

(3) Supplementary Figures. The order of appearance in the main text is occasionally inconsistent.

This was modified, thanks

(4) Line 132. Add a reference to the registration toolbox used (elastix). A brief description of the affine transformation would also be helpful, either here or in the Methods section (p. 27).

We have added reference to Ntatsis et al. 2023 and described affine transformation in the main text (lines 133-135):

Firstly, we estimate the spatial transformation between s0 and s1 using affine image registration (i.e. allowing shifting, rotation, scaling and shearing, see Fig. 2B, the transformation is denoted as T).

(5) Lines 147-151. If this method is adapted from another work, please cite the source.

Computing the intersection over union of two ROIs for tracking is a widely established and intuitive method used across numerous studies, representing standard practice rather than requiring specific citation. We have however included the reference to the paper describing the algorithm we use to solve the linear sum assignment problem used for matching neurons across a pair of consecutive days (Crouse 2016).

(6) Line 218. "classical" or automatic?

We meant “classical” in the sense of widely used.

(7) Lines 220-231. Did the authors find significant variability of successfully tracked neurons across mice? While the data for successfully tracked cells is reported (Figure 5B), the proportions are not. Could differences in neuron dropout across days and mice affect the analysis of neuronal activity statistics?

We thank the reviewer for raising this important point. We computed the fraction of successfully tracked cells in our dataset and found substantial variability:

Cells detected on day 0: [607, 1849, 2190, 1988, 1316, 2138]

Proportion successfully tracked: [0.47, 0.20, 0.36, 0.37, 0.41, 0.19]

Notably, the number of cells detected on the first day varies considerably (607–2138 cells). There appears to be a trend whereby datasets with fewer initially detected cells show higher tracking success rates, potentially because only highly active cells are identified in these cases.

To draw more definitive conclusions about the proportion of active cells and tracking dropout rates, we would require activity-independent cell detection methods (such as Cellpose applied to isosbestic 830 nm fluorescence, or ideally a pan-neuronal marker in a separate channel, e.g., tdTomato). We have incorporated the tracking success proportions into the revised manuscript.

(8) Line 260. Please briefly explain, here or in the Methods, the rationale for using data from only 3 mice (rather than all 6) for evaluating tracking performance.

We used three mice for this analysis due to the labor-intensive nature of manually annotating 64 ROIs across several days. Given the time constraints of this manual process, we determined that three subjects would provide adequate data to reliably assess tracking performance.

(9) Line 277. Consider clarifying or rephrasing the phrase "across progressively shorter time intervals"? Do you mean across consecutive days?

This has been rephrased as follows:

Additionally, to assess tracking performance over time, we quantified the proportion of reconstructed ground truth tracks over progressively longer time intervals (first two days, first three days etc. ‘Prop. correct’ in Fig. 4C-F, see Methods). This allowed us to understand how tracking accuracy depends on the number of successive sessions, as well as at which time points the algorithm might fail to successfully track cells.

(10) Line 306. "we also provide additional resources and documentation". Please add a reference or link.

Done, thanks

Track2p

(11) Lines 342-344. Specify that the raster plots refer to one example mouse, not the entire sample.

Done, thanks.

(12) Lines 996-1002. Please confirm whether only successfully tracked neurons were used to compute the Pearson correlations between all pairs.

Yes of course, this only applies to tracked neurons as it is impossible to compute this for non-tracked pairs.

(13) Line 1003. Add a reference to scikit-learn.

Reference was added to:

Pedregosa, F., Varoquaux, G., Gramfort, A., Michel, V., Thirion, B., Grisel, O., Blondel, M., Prettenhofer, P., Weiss, R., Dubourg, V., Vanderplas, J., Passos, A., Cournapeau, D., Brucher, M., Perrot, M., & Duchesnay, E. (2011). Scikit-learn: Machine Learning in Python. Journal of Machine Learning Research, 12, 2825–2830.

(14) Typos.Correct spacing between numeric values and units.

We did not find many typos regarding spacing between the numerical value and the unit symbol (degrees and percent should not be spaced right?).

**Reviewer #3 (Recommendations for the authors):**
The font size in many of the figures is too small. For example, it is difficult to follow individual ROIs in Figure S3.

Figure font size has been increased, thanks. In Figure S3 there might have been a misunderstanding, since the three FOV images do not correspond to the FOV of the same mouse across three days but rather to the first recording for each of the three mice used in evaluation (the ROIs can thus not be followed across images since they correspond to a different mouse). To avoid confusion we have labelled each of the FOV images with the corresponding mouse identifier (same as in Fig. 4 and 5).